# Enteric nervous system regeneration and functional cure of experimental digestive Chagas disease with trypanocidal chemotherapy

Archie A. Khan[1], Harry C. Langston [1], Louis Walsh[1], Rebecca Roscoe[1], Shiromani Jayawardhana[1], Amanda Fortes Francisco [1], Martin C. Taylor [1], Conor J. McCann [2], John M. Kelly [1] & Michael D. Lewis [1,3] ✉

Digestive Chagas disease (DCD) is an enteric neuropathy caused by *Trypanosoma cruzi* infection. There is a lack of evidence on the mechanism of pathogenesis and rationales for treatment. We used a female C3H/HeN mouse model that recapitulates key clinical manifestations to study how infection dynamics shape DCD pathology and the impact of treatment with the front-line, anti-parasitic drug benznidazole. Curative treatment 6 weeks post-infection resulted in sustained recovery of gastrointestinal transit function, whereas treatment failure led to infection relapse and gradual return of DCD symptoms. Neuro/immune gene expression patterns shifted from chronic inflammation to a tissue repair profile after cure, accompanied by increased cellular proliferation, glial cell marker expression and recovery of neuronal density in the myenteric plexus. Delaying treatment until 24 weeks post-infection led to partial reversal of DCD, suggesting the accumulation of permanent tissue damage over the course of chronic infection. Our study shows that murine DCD pathogenesis is sustained by chronic *T. cruzi* infection and is not an inevitable consequence of acute stage denervation. The risk of irreversible enteric neuromuscular tissue damage and dysfunction developing highlights the importance of prompt diagnosis and treatment. These findings support the concept of treating asymptomatic, *T. cruzi*-infected individuals with benznidazole to prevent DCD development.

Chagas disease (CD), or American trypanosomiasis, is a neglected tropical disease with a prevalence of 6.5 million cases, a burden of 10,000 deaths per year, 275,000 DALYs and economic costs reaching US$7 billion per year[1,2]. The large majority of cases occur in endemic regions of Latin America, but there is a clear long-term trend of globalisation[3–5]. Chagas Disease is caused by *Trypanosoma cruzi*, a

protozoan parasite, which is primarily transmitted to humans by blood-feeding insect vectors (triatomine bugs), but it can also be acquired congenitally or from contaminated blood transfusions, organ transplants and foodstuffs[6]. Anti-parasitic treatment is limited to the nitroheterocyclic drugs, nifurtimox and benznidazole (BZ). Both have long dosing schedules and can cause significant toxicity[7,8]. Daily BZ

[1]Department of Infection Biology, London School of Hygiene and Tropical Medicine, Keppel Street, WC1E 7HT London, UK. [2]Stem Cells and Regenerative Medicine, University College London, Great Ormond Street Institute of Child Health, London, UK. [3]Division of Biomedical Sciences, Warwick Medical School, University of Warwick, CV4 7AJ Coventry, UK. ✉e-mail: michael.d.lewis@warwick.ac.uk

treatment for 60 days is the current standard of care because side effects are considered less severe than for nifurtimox. Recent trial data show that reducing the duration of treatment to 2 weeks may be justified[9].

Upon transmission, *T. cruzi* invades target cells of diverse types and begins an approximately weekly cycle of replication, host cell lysis and dissemination. In most cases, adaptive immunity suppresses parasite numbers to very low levels; sterile clearance is considered rare[10,11]. Clinical manifestations affecting the heart and/or gastro-intestinal (GI) tract develop in around one-third of chronically infected people. Benznidazole treatment is recommended for all acute, congenital and immunosuppression-related reactivation cases, as well as chronic infections in children and women of childbearing age[12]. However, the evidence for the efficacy of BZ in terms of chronic disease progression and outcomes is limited. Treatment showed no significant benefit compared to placebo in terms of preventing death or disease progression in patients who already had symptomatic cardiac Chagas disease[8]. Clinical and pre-clinical data on the impact of treatment on digestive Chagas disease (DCD) outcomes are lacking.

DCD is an enteric neuropathy characterised by progressive dilatation and dysfunction of sections of the GI tract[13,14]. Symptoms include achalasia, abdominal pain, constipation and faecaloma. Eventually, in some cases, massive organ dilatation results in megasyndromes, usually of the colon and/or oesophagus. Dilatation is associated with loss of enteric neurons, leading to peristaltic paralysis and smooth muscle hypertrophy. Options for DCD management are limited to palliative and surgical interventions[15], often implemented in emergency scenarios late in the disease course, with significant mortality risk[16].

DCD is thought to stem from collateral damage to enteric neurons caused by anti-parasitic inflammatory immune responses in the muscle wall of the affected region of the GI tract[17]. Beyond this, the mechanism and kinetics of denervation, and therefore a rationale for treatment, are poorly defined. The inability to detect gut-resident parasites in chronic infections supported a model of acute phase damage unmasked by further ageing-related denervation[18]. Molecular detection of *T. cruzi* DNA and inflammatory infiltrates in post-mortem and biopsy studies of human DCD circumstantially suggests that chronic parasite persistence may contribute to disease development[19–27]. These data from late and terminal disease states are difficult to interpret with respect to relationships between pathogenesis and infection load or its distribution over time. Experimental bioluminescence imaging and tissue PCR studies in mice showed that the GI tract is a major long-term reservoir of infection with diverse *T. cruzi* strains[28–36]. This led to the development of a robust mouse model of DCD, which features significantly delayed GI transit associated with co-localised parasite persistence and enteric neuronal lesions in the wall of the large intestine[37]. Here, we utilised this model to formally test the hypothesis that a BZ-mediated cure of *T. cruzi* infection can either prevent DCD, or reduce its severity.

## Results

### Benznidazole-mediated cure of *T. cruzi* infection in the experimental DCD model
We have developed a model of chronic DCD based on female C3H/HeN mice infected with bioluminescent TcI-JR parasites[37] (Fig. 1). Subsets of mice were treated with BZ or vehicle at 6 weeks post-infection (wpi) (Fig. 1a, b). At this time, parasite loads are already in sharp decline as a result of adaptive immunity driving the transition from acute to chronic infection. In vivo, bioluminescence imaging (BLI) showed that infected mice administered with the vehicle alone, hereon "untreated" infected mice, transitioned to a stable, low-level chronic infection (Fig. 1b, c, Supplementary Fig. 1a). In contrast, parasite loads in BZ-treated mice became undetectable by in vivo and post-mortem ex vivo BLI (Fig. 1b, c, g, Supplementary Fig. 1). This was corroborated by splenomegaly, low body weight, increased caecum weight and loss of GI mesenteric tissue at 36 wpi in the untreated infected group. In all

cases, these read-outs reversed closer to the uninfected control baseline after curative BZ treatment (Fig. 1d, Supplementary Fig. 2). In vivo and ex vivo BLI identified a subset of BZ-treated mice ($n = 13/27$, 48%) in which treatment failed and the infection relapsed (Fig. 1b, c, e). Of these, 7 (26%) infections were only detected by post-mortem ex vivo imaging of internal organs (Fig. 1e). Retrospective comparison of body weights and parasite loads showed there was no difference at the start of treatment between animals that were subsequently cured and those in which treatment failed (Supplementary Fig. 3a–c). Relapse infections were significantly less disseminated amongst organs and tissue types than untreated infections (Fig. 1f, Supplementary Data 1). The most common sites of relapse were the large intestine (8/13, 62%), GI mesentery (7/13, 54%) and heart (6/13, 46%). Of note, in the context of cardiac Chagas disease, relapse infections localised at a significantly lower rate to the heart, which was a site of frequent, high-intensity parasitism in the untreated group (17/18, 94%) (Fisher's exact test $P = 0.0041$; Fig. 1f, g, Supplementary Data 1). However, given the capacity of *T. cruzi* trypomastigotes to periodically traffic within and between organs, these snap-shot parasite distribution profiles might not fully reflect the spatio-temporal dynamism of relapse infections. Overall, our findings show that BZ treatment at 6 wpi achieved 51.9% sustained parasite clearance, here considered a parasitological cure, with infection relapse cases most often localised to the large intestine.

### Benznidazole treatment restores normal GI transit function
In our DCD model, there is a highly significant delay in GI transit time in infected mice compared to uninfected controls[37] (Fig. 2a, b). Benznidazole chemotherapy rapidly reversed the transit delay phenotype to the uninfected control baseline (Fig. 2a, b). Immediately prior to initiation of treatment, the mean GI transit time was 184 min in untreated infected mice compared to 103 min in uninfected controls (Fig. 2a). In the untreated infection group, the delay remained significant, although it initially eased, in line with immune-mediated parasite load reduction, and then gradually worsened as the chronic phase progressed (Fig. 2a, b). Curative BZ treatment led to permanent restoration of normal transit times. Importantly, relapse infections were associated with the return of a significant transit delay, but this remained less severe than for the untreated infected group (Fig. 2a, b). There was no correlation between the level of relapse and transit delay in individual mice at discrete time points (Supplementary Fig. 3d), but over time the average levels followed similar, worsening trajectories (Supplementary Fig. 3e). At the experiment end-point (36 wpi), we analysed faecal retention in the colon after a period of fasting. This showed a clear constipation phenotype associated with a significantly increased faecal pellet number and weight in both untreated and relapsed infections (Fig. 2c, d). This was alleviated in BZ-cured mice to the point that they were not significantly different from uninfected control mice (Fig. 2c–e). We also observed significant normalisation of caecum weight in cured mice (Supplementary Fig. 2c).

To further investigate differences in colonic motility independent of connections to the CNS, we evaluated ex vivo basal contractility of colon tissue samples (Fig. 2f). Electrophysiological data showed that the contractility frequency in colons from untreated infected mice was significantly reduced compared to uninfected controls (Fig. 2g). Colons from BZ-treated infected mice displayed restoration of basal contractile frequency in both the BZ-cured and BZ-relapsed groups, to levels significantly higher than the untreated infected group and not significantly different from the healthy, uninfected controls (Fig. 2f, g). The amplitude of basal contractions was not significantly changed by infection or treatment, however, a trend of reduced amplitude was observed in colons from the untreated and relapse infections (Fig. 2h). Although it was only possible to assess a small subset of samples by contractility analysis, the data were broadly consistent with the total GI transit time phenotypes and suggest that BZ-mediated suppression or cure of infection supports enteric nervous system (ENS) functional recovery.

## GI transit recovery is associated with re-innervation of myenteric plexus ganglia

Given the relevance of denervation to human DCD, we next evaluated the impact of infection and BZ-mediated treatment on the ENS. At 3 weeks post-infection (i.e. 3 weeks pre-treatment), we observed atypical expression patterns for standard markers of enteric neurons and glial cells in the colonic muscularis propria, including clear loss of

discrete Hu+ neuronal cell bodies, hereon "Hu+ soma" (Fig. 3a). TUNEL staining inside myenteric plexus ganglia indicated that acute *T. cruzi* infection led to DNA damage characteristic of apoptosis in the ENS (Fig. 3a). We also detected activation of the apoptotic executioner caspase-3 inside myenteric ganglia (Supplementary Fig. 4a). *T. cruzi*-infected mice exhibited a spectrum of ENS damage by the time BZ treatment was initiated (6 wpi) and this continued in the untreated

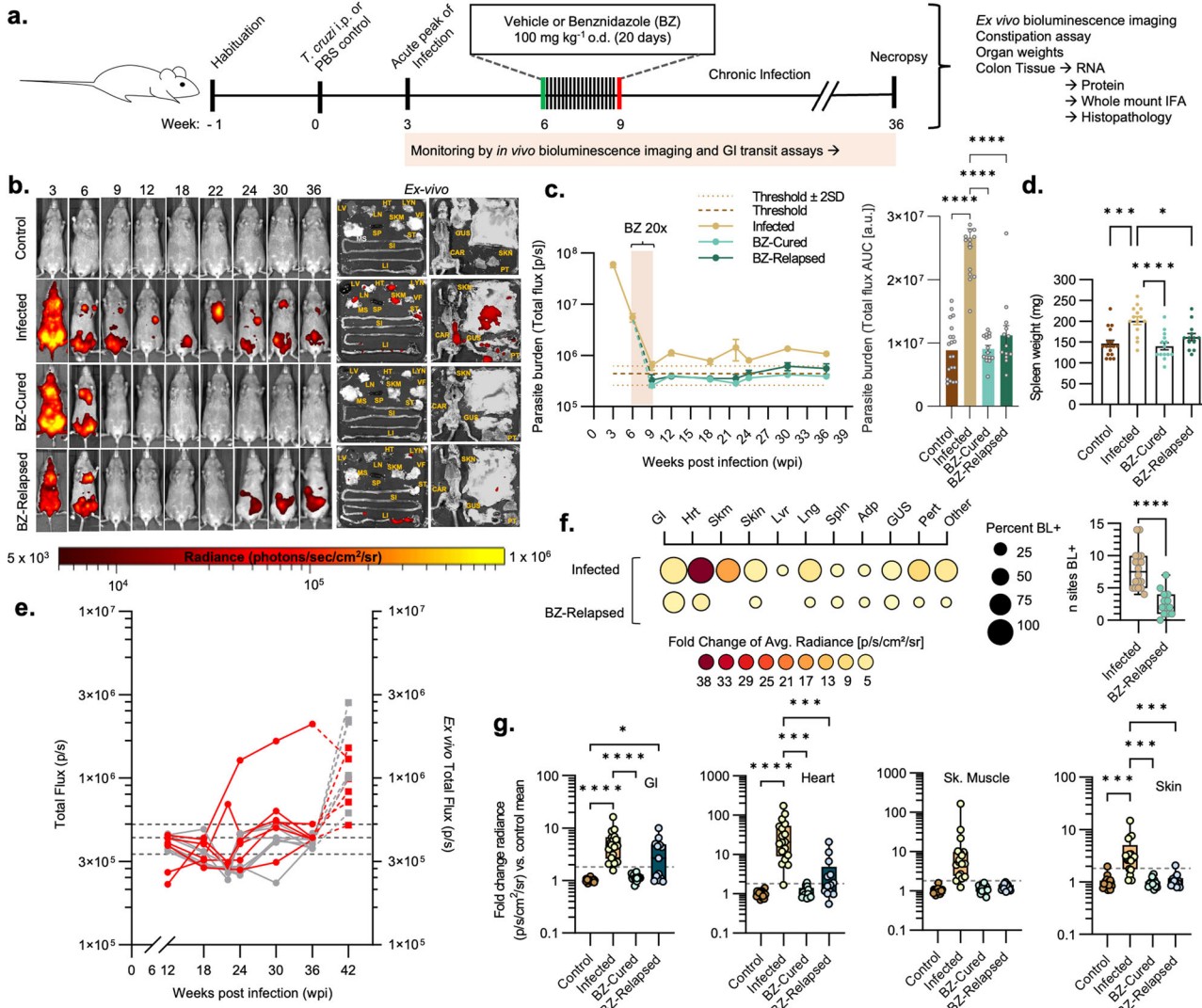

**Fig. 1 | Evaluation of benznidazole treatment outcomes in murine digestive Chagas disease. a** Schematic representation of the experiment. **b** Representative in vivo bioluminescence (BL) images of female C3H/HeN mice that were uninfected (control), infected with TcI-JR and (i) vehicle-administered (infected), or treated with benznidazole at 6 wpi and (ii) assessed as parasitologically cured (BZ-Cured), or (iii) assessed as treatment failure (BZ-Relapsed). Ex vivo images show bioluminescence in liver (LV), lymph nodes (LYN) lungs (LN), gut mesenteries (MS), heart (HT), spleen (SP), skeletal muscle (SKM), visceral fat (VF), stomach (ST), small intestine (SI), large intestine (LI), skin (SK), carcass (CAR), genito-urinary system (GUS), and peritoneum (PT). BL intensity expressed using radiance (p/s/cm²/sr) pseudocolour heat maps. **c** In vivo BL profiles of infected (n = 18, except n = 10 at 9 and 22 wpi, n = 47 at 6 wpi and n = 50 at 3 wpi), BZ-Cured (n = 14, except n = 15 at 12 and 18 wpi, n = 7 at 9 wpi and n = 6 at 22 wpi) and BZ-Relapsed (n = 13, except n = 6 at 9 and 22 wpi) mice Dashed lines show uninfected control auto-luminescence-based thresholds. Bar plots show cumulative parasite burdens based on the area under the curve of the line plots (control n = 20, infected n = 18, BZ-Cured n = 15, BZ-Relapsed n = 13). **d** Spleen weights of control (n = 17), infected (n = 14), BZ-Cured (n = 16) and BZ- Relapsed (n = 11) groups. **e** Post-treatment in vivo BL infection profiles (left y-axis; round data points, full lines) and ex vivo bioluminescence at

36 wpi (right y-axis, sum of all ROIs, square data points, dashed lines) for individual BZ-Relapsed mice. Data in red indicate relapses detected during in vivo infection and grey at ex vivo stage. Thresholds as in (**c**). **f** Mean tissue-specific infection intensities in untreated and relapsed infections. Circle size indicates the percentage of individual animals with BL-positive (BL+) signal for each sample type; colour indicates infection intensity (fold change in ex vivo BL vs uninfected controls). Infection dissemination box plot shows the number of BL+ tissue types per mouse. Infected n = 18 (except skin n = 17), BZ-Relapsed n = 13 mice; two independent experiments. **g** Tissue-specific infection intensities for the GI tract, heart, skeletal muscle and skin of control (n = 19; except skin n = 16), infected (n = 18; except skin n = 17), BZ-Cured (n = 14) and BZ-Relapsed (n = 13) mice over two independent experiments. Data expressed as mean fold change in radiance vs. uninfected mean. Threshold line is the mean for an internal control (empty) region of interest. Box plots show the median with minimum and maximum values as whiskers; the bounds of the box show IQR. All other data are expressed as mean ± SEM. Statistical significance was tested using a two-sided t-test or one-way ANOVA with Tukey's HSD test. Only significant differences are annotated: *P < 0.05, **P < 0.01, ***P < 0.001, **** P < 0.0001.

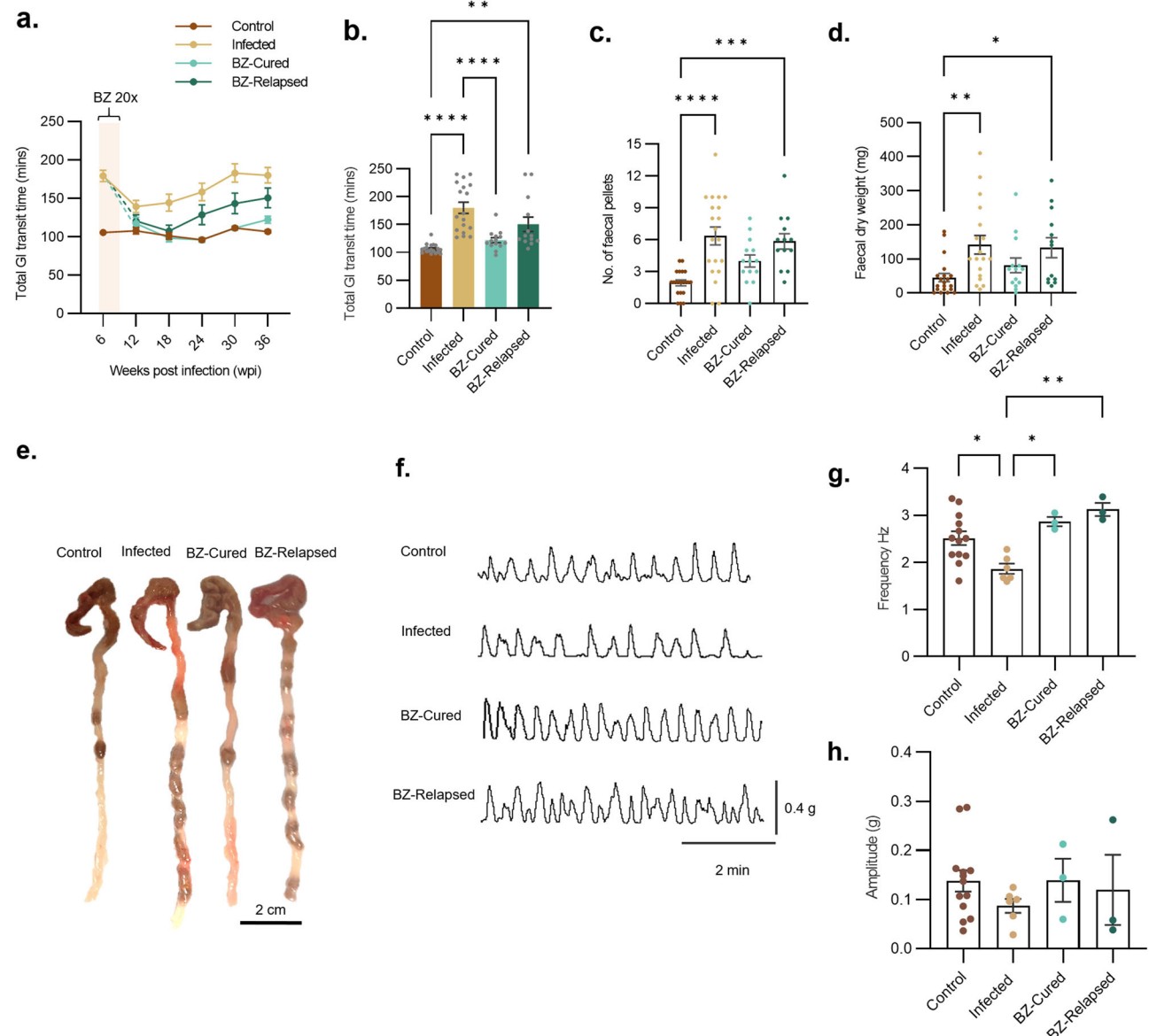

**Fig. 2 | Early cure of infection leads to durable restoration of normal GI transit function. a** Line plots show total GI transit time for control ($n = 20$), infected ($n = 18$, except $n = 46$ at 6 wpi), benznidazole (BZ) treated and cured (BZ-Cured; $n = 14$, except $n = 15$ at 12 and 18 wpi) and BZ-treated and relapsed (BZ-Relapsed; $n = 13$) C3H/HeN mice against weeks post-infection (wpi). Cream bar shows the BZ treatment window (6–9 wpi). **b** Bar plots show individual animal data for end-point (36 wpi) total GI transit time; control ($n = 19$), infected ($n = 18$), BZ-Cured ($n = 14$) and BZ-relapsed ($n = 13$). Control vs Infected $p < 0.0001$; Control vs BZ-Relapsed $P = 0.002$; Infected vs BZ-Cured $p < 0.0001$. **c, d** Bar plots show post-mortem number of faecal pellets (**c**) and dry faecal pellet weight (**d**, sum of all pellets) in the colon of control ($n = 20$), infected ($n = 18$), BZ-Cured ($n = 14$) and BZ-Relapsed ($n = 13$) mice after 4 h' fasting. For (**c**) Control vs Infected $p < 0.0001$; Control vs BZ-Relapsed $P = 0.0006$. For (**d**) Control vs Infected $P = 0.009$; Control vs BZ-Relapsed $P = 0.04$. **e** Images of control, infected, BZ-Cured and BZ-Relapsed mouse large intestine and retained faecal pellets after 4 h' fasting at 36 wpi. Scale bar = 2 cm. **f** Representative proximal colon basal contractile traces from organ bath contractility assay for each experimental group. **g, h** Bar plots show basal contraction frequency and amplitude respectively of control ($n = 13$), infected ($n = 6$), BZ-Cured ($n = 3$) and BZ-Relapsed ($n = 3$) mice. For (**g**) Control vs Infected $P = 0.023$; Infected vs BZ-Relapsed $P = 0.002$; Infected vs BZ-Cured $P = 0.014$. Data are expressed as mean ± SEM. Statistical significance was tested using one-way ANOVA followed by Tukey's HSD test. Only significant differences are annotated: *$P < 0.05$, **$P < 0.01$, ***$P < 0.001$, **** $P < 0.0001$.

infections up to the end-point of the experiment (36 wpi). Specifically, we observed further evidence of neuronal cell death in the form of denervated ganglia, loss of typical Hu+ soma morphology and pyknotic nuclei (Fig. 3b, c).

At the pre-treatment baseline, 6 wpi, the number of Hu+ soma in the myenteric plexus of infected mice was significantly reduced compared to uninfected controls, by 70% and 77% in the proximal and distal colon respectively (Fig. 3d). At 12 wpi (3 weeks after BZ withdrawal), the neuron density remained at these reduced levels in untreated infected controls, but there was evidence of a recovery

trend for Hu+ neuron morphology and numbers in mice that had been treated with BZ, which by this stage were 49% lower than the normal density (Supplementary Fig. 4b, 4c). At the end of the treatment follow-up period, 36 wpi (27 weeks after BZ withdrawal), myenteric neuronal density had declined in untreated infections to 85% and 71% loss in the proximal and distal colon respectively (Fig. 3c, d). Benznidazole-mediated cure of infection led to the recovery of neuron morphology and numbers, with only 32% and 16% less than the uninfected control means in the proximal and distal myenteric plexus respectively. Denervation in relapsed mice was significant, but of lower

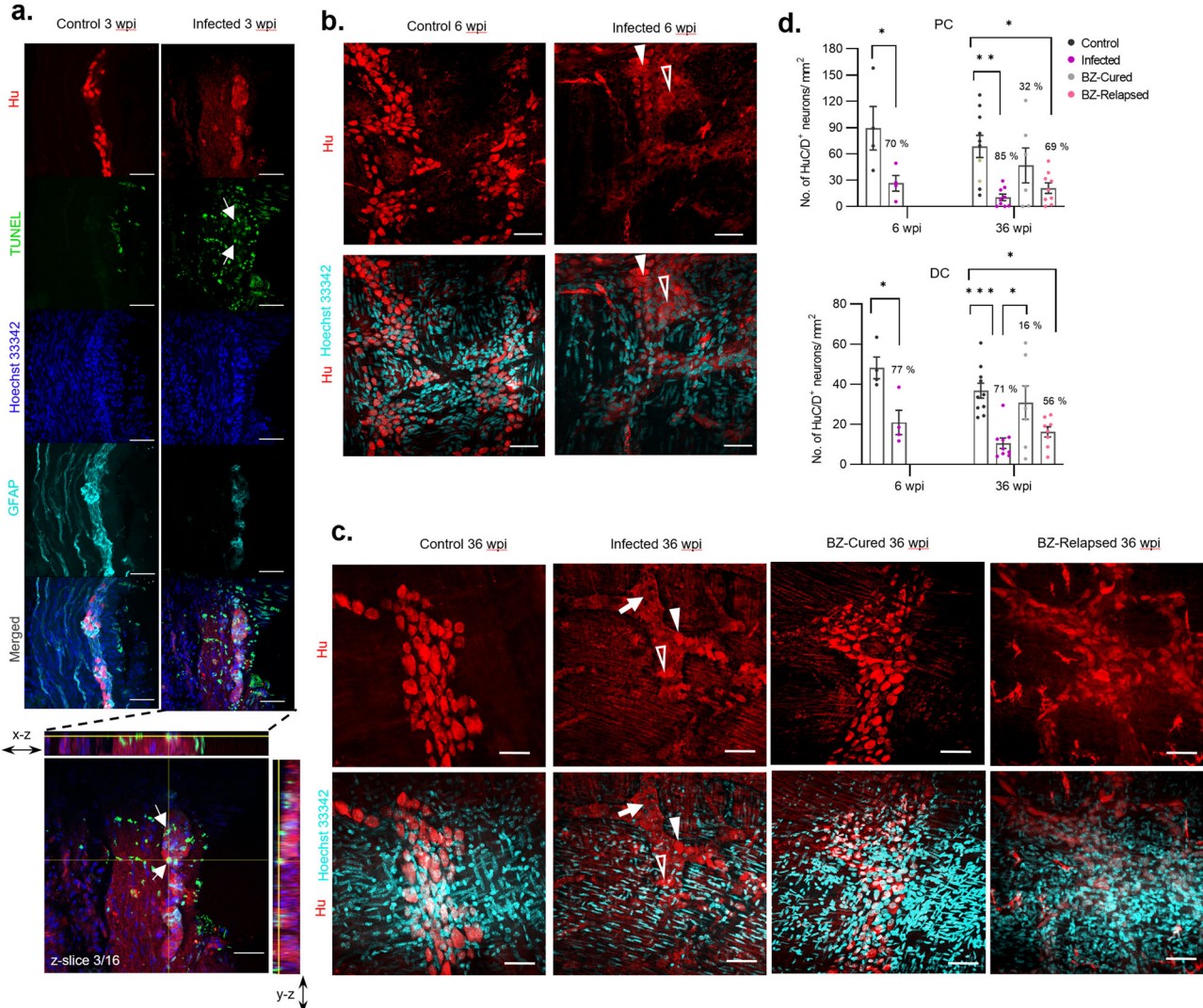

**Fig. 3 | Enteric nervous system cellular damage, death and recovery in the colon myenteric plexus. a** Representative, compressed z-stack images of transverse colon sections from C3H/HeN mice after 3 weeks of *T. cruzi* infection and age-matched uninfected controls. Immunofluorescence analysis shows acute cellular damage of Hu+ neuronal cell bodies (soma, red) and GFAP+ glial cells (cyan). Apoptotic DNA damage is marked by TUNEL staining (green) co-localised with Hoechst 33342 DNA stain (blue), with intra-ganglionic events highlighted (white arrows). Expanded image and orthogonal views of a single z-slice from the merged image of the infected colon showing co-localisation of TUNEL stain with areas of diminished neuronal Hu protein expression (white arrows) in close proximity to GFAP+ glial cells. Orthogonal view of z-planes of the image across x- and y-axis are marked by yellow lines. **b**, **c** Representative images of whole-mount colons at 6 and 36 weeks post-infection (wpi) respectively from control and infected C3H/HeN mice and additionally in (**c**) from benznidazole (BZ) treated and cured (BZ-Cured), and BZ-treated and relapsed (BZ-Relapsed) infections. Immunofluorescent labelling shows changes of Hu+ soma (red) with and without DNA stain (Hoechst 33348, cyan) in the myenteric plexus. White arrow indicates the area of neuropathy lacking defined soma, with weak Hu expression and pyknotic nuclei; filled arrowheads indicate intact soma morphology with typical Hu expression and pyknotic nuclei; empty arrowheads indicate irregular soma morphology with Hu expression and intact nuclei. All micrographs are representative images of two independent experiments. **d** Bar plots show number of Hu+ neurons in proximal (PC) and distal colon (DC) before (6 wpi) and after BZ treatment (36 wpi) of control (*n* = 4 at 6 wpi, *n* = 10 at 36 wpi), infected (*n* = 4 at 6 wpi, *n* = 9 at 36 wpi), BZ-Cured (*n* = 6) and BZ-Relapsed (*n* = 9) groups. Data are expressed as mean ± SEM. Statistical significance was tested using one-way ANOVA followed by Tukey's HSD test. Only significant differences are annotated: *P < 0.05, **P < 0.01, ***P < 0.001. All confocal images were taken at 400× magnification, scale bar = 50 μm.

magnitude (69% proximal, 56% distal) than for untreated infected mice (Fig. 3d), in line with their intermediate transit delay phenotype (Fig. 2a–d). We observed a morphologically heterogeneous population of Hu+ myenteric neuronal bodies in colon samples from BZ-cured mice (Fig. 3c, Supplementary Fig. 4d). A subset of these neurons resembled those seen in healthy control ganglia, while another subset appeared atypically smaller and rounder, with weaker anti-Hu reactivity. These were commonly present in the same ganglion as neurons with normal soma morphology and neighbouring healthy control-like myenteric ganglia (Supplementary Fig. 4d).

Together, these data show that reversal of the DCD transit time and constipation phenotypes after cure of *T. cruzi* infection is

associated with substantial recovery of myenteric neuron density, particularly in the distal colon. In mice in which treatment failed and infection relapsed, transit time delay returned, but not to the levels observed in untreated infections, with an intermediate recovery of myenteric innervation.

## Distinct gene expression profiles for chronic, relapsed and cured infections
To further investigate how the balance of infection and host immunity impacts the ENS during *T. cruzi* infection, we performed a Nanostring multiplex analysis of host gene expression, focussing on immune response (*n* = 491) and ENS (*n* = 17) genes. In mice with untreated

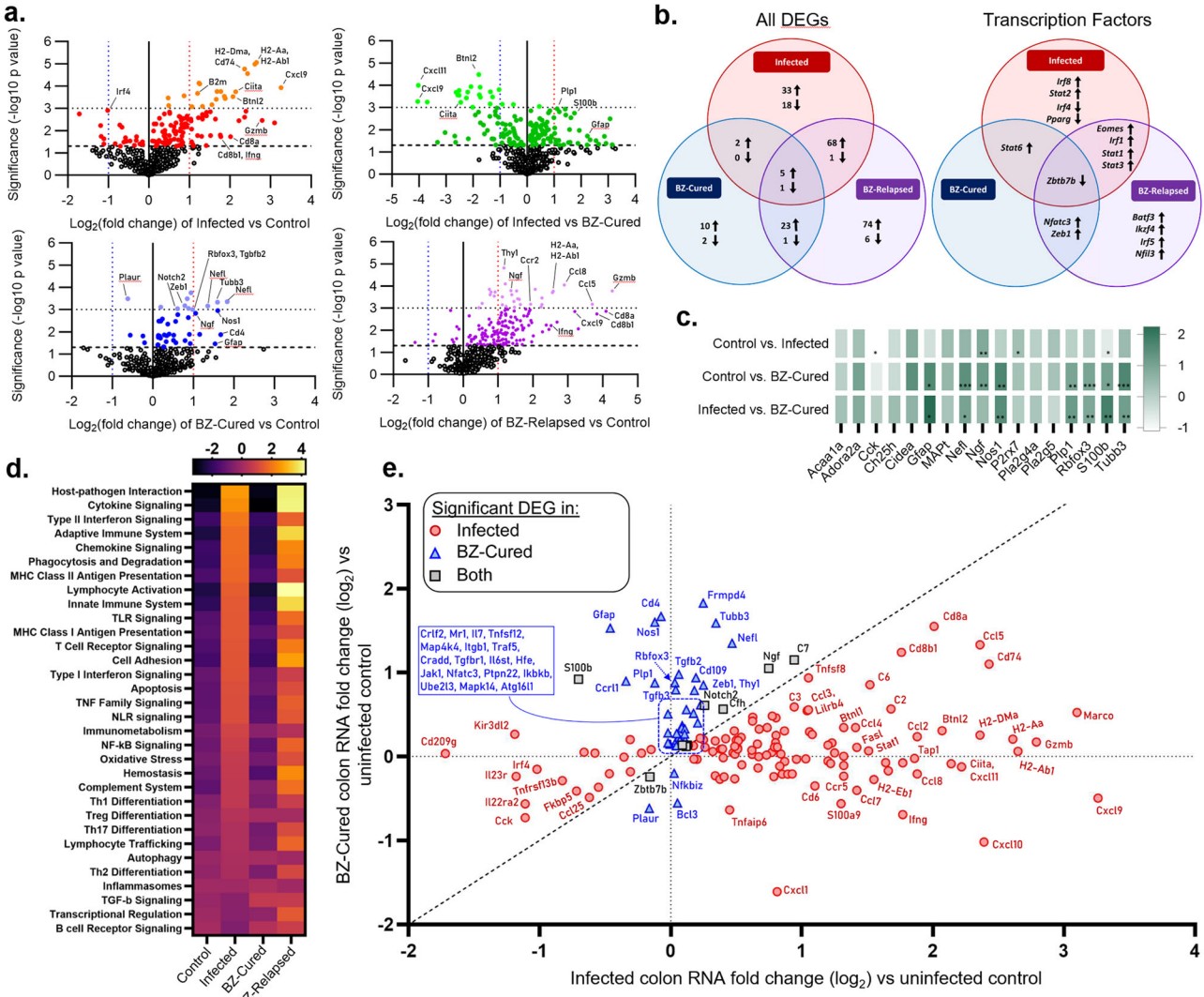

**Fig. 4 | Enteric neuro/immuno gene expression profiles associated with *T. cruzi* infection and benznidazole treatment outcomes. a** Volcano plots of the log$_2$-transformed fold change and significance (−log$_{10}$ *P*-value) of differentially expressed genes (DEGs) in colon tissue from infected (red/orange shades), benznidazole (BZ) treated and cured (BZ-Cured; blue shades) and BZ-treated and relapsed (BZ-Relapsed; purple shades) vs. uninfected control C3H/HeN mice; and BZ-Cured vs. infected (green shades) mice. Darker coloured shaded dots indicate statistical significance of *P* < 0.05 (above dashed horizontal line) and lighter shaded dots *P* < 0.001 (above dotted horizontal line). **b** Venn diagrams show same-direction differentially expressing genes (DEG) shared between infected, BZ-Cured and BZ-Relapsed for all genes analysed (left, *n* shows number of genes) and transcription factors (right) (threshold of *P* < 0.05 vs uninfected controls in at least one group). Arrows indicate up- and down-regulation. **c**, Relative change in neuro-glial

genes between indicated experimental groups; colour intensity indicates fold change (log$_2$) expression vs uninfected controls. **d** Signalling pathway scores for each group. **e** Comparison of directionality and extent of gene expression change in chronically infected and BZ-Cured mice vs. uninfected controls (*n* = 163 genes that are significant DEGs in at least one group). Red circles are DEGs specific to chronic, untreated infections, blue triangles are DEGs specific to the BZ-Cured mice and grey squares are DEGs shared by both groups. Diagonal dashed line is the line of equivalence. Vertical and horizontal dashed lines indicate the position of genes with identical expression levels as uninfected controls in chronic infection and BZ-Cured mice respectively. Infected and uninfected controls *n* = 6, BZ-Cured and BZ-Relapsed *n* = 3. Statistical significance was determined by a 2-tailed, unpaired Student's *t*-test for each gene. Only significant differences are annotated: *\*P* < 0.05, *\*\*P* < 0.01, *\*\*\*P* < 0.001, *\*\*\*\* P* < 0.0001.

chronic infections (36 wpi), there were 128 significantly differentially expressed genes (DEGs) compared to uninfected controls in colon tissue, of which 108 were up- and 20 down-regulated (Fig. 4a, b; Supplementary Data 2). These evidenced a type 1-polarised inflammatory response involving class I and class II antigen presentation (e.g. *Ciita, Tap1, Psmb9, B2m; H2-Aa, H2-Ab1, Cd74*), cytokines/chemokines (e.g. *Ifng, Il21, Tnf; Cxcl9, Cxcl10, Ccl5*), transcription factors (e.g. *Stat1,−2,−3,−6, Irf1,−8*), cytotoxic lymphocyte markers and effectors (*Cd8a, Cd8b1, Cd226, Gzmb, Fasl, Prf1*), complement factors (*C2, C6, C4a, C3, C7, C1qb*) and Fc receptors (*Fcgr1,−4,−3, Fcer1a,−1g*). There was also evidence of significant dysregulation at the pathway level for antigen presentation, interferon signalling, apoptosis and phagocytosis (Fig. 4d, Supplementary Fig. 5a). Consistent with the enduring

capacity of *T. cruzi* to survive in this inflammatory environment and the need to prevent excessive GI tissue damage, the upregulated DEG set included a diverse range of immuno-inhibitory mediators: *Btnl1, Btnl2, Cd274* (PD-L1), *Socs1, Lilrb4, Lilrb3, Lair1, Tnfaip3, Serping1* (Fig. 4a, e, Supplementary Data 2).

Next, we evaluated the impact of BZ treatment success and failure on host gene expression. Relapsed infections were associated with a larger DEG set (*n* = 179 vs uninfected controls) than untreated infections. The majority, 58.6%, were same-direction DEGs as chronic infections, but the data also revealed relapse-specific changes (Fig. 4b, Supplementary Fig. 6). Notably, these included stronger upregulation of 54 genes, including *Cd8a, Cd8b1, Ccl5, Cxcl1, Gzmb* and *Ifng*, and unique upregulation of cytotoxic effectors *Gzma* and *Fas*, leukocyte

markers *Cd4*, *Cd7* and *Cd27*, transcription factors (*Batf3*, *Ikzf4*, *Irf5*, *Nfil3*) and the components of integrins α4β1 (VLA-4), α5β1 (VLA-5), αLβ2 (LFA-1), αMβ2 (MAC-1). Given the lower and less disseminated parasite loads seen in these animals (Fig. 1), this broader gene expression profile is consistent with enhanced control of *T. cruzi*, enduring for months after non-curative treatment.

Colon tissue from BZ-cured mice had gene transcript abundances equivalent to uninfected controls for 119 (93%) of the 128 DEGs that were identified in untreated chronic infections, indicating near complete reversion to homoeostasis (Fig. 4). At the pathway level, there were no significant differences between cured and uninfected groups (Supplementary Fig. 5a). Nevertheless, cured mice did have a distinct profile compared to uninfected control mice, comprising 44 DEGs (40 up- and 4 down-regulated genes) (Fig. 4b, c, e). Of note, the most upregulated gene set was enriched for neuronal markers (*Rbfox3* (NeuN), *Nefl*, *Tubb3*), enteric glial cells (EGCs) (*S100b*, *Gfap*, *Plp1*), genes associated with neural development (*Ngf*, *Frmpd4*) and neurotransmission (*Nos1*). The EGC marker *S100b* was the only switched-direction DEG, with reduced expression in chronic infections and increased expression in cured mice (Fig. 4c, e). Furthermore, multiple genes involved in tissue repair and regeneration were also highly significant DEGs in cured mice, including *Notch2*, *Tgfb2*, *Tgfbr1*, *Tgfb3* and *Zeb1* (Fig. 4a, e, Supplementary Data 2). Cure of *T. cruzi* infection therefore results in dissipation of the chronic inflammatory environment in the colon and induction of a tissue repair programme that encompasses key components of the ENS.

### Neuro-glial impact of benznidazole treatment

Nitrergic neuronal inhibitory signalling is critical for homoeostatic control of peristalsis and its disruption is linked to a range of enteric neuropathies, including human DCD[38]. Evidence for altered *Nos1* gene expression in our experimental model[37] (Fig. 4c, e) led us to analyse the expression of the corresponding protein (neuronal NOS, nNOS) in myenteric neurons from the proximal and distal colon using immunofluorescence. The number of distal colon nNOS⁺ neurons was significantly reduced in untreated and relapsed infections compared to uninfected controls (Fig. 5a, b). In BZ-cured animals, nNOS⁺ neuron density remained lower on average than uninfected controls, but the difference was not statistically significant (Fig. 5a, b).

The RNA-based analyses strongly implicated EGCs in post-cure tissue repair (Fig. 4c, e, Supplementary Fig. 5b), so these were further analysed at the protein level. Immunofluorescence analysis of neuro-glial network morphology (Fig. 5c) clearly showed EGCs expressing glial fibrillary acidic protein (GFAP⁺) wrapped around neurons (neuron-specific β-tubulin, TuJ1⁺) in the myenteric plexus for all groups. Networks of GFAP⁺ EGCs in samples from untreated infections showed the weakest staining intensity and contained fragmented GFAP aggregates and patches in the plexus compared to controls, cured and relapsed mice. Conversely, GFAP expression in samples from BZ-treated animals exhibited control-like glial morphology with a dense network, which was also preserved in relapsed infections (Fig. 5c).

Western blot analysis of GFAP protein expression revealed bands at approximately 100 kDa, larger than reference brain tissue control (50 kDa; Supplementary Fig. 7), which may reflect dimer formation[39]. A secondary band was observed in both the untreated and relapsed infection groups at approximately 80 kDa, indicating that altered GFAP protein structure might be a feature of DCD pathology. Benznidazole treatment induced a doubling of GFAP protein abundance compared to controls and the disappearance of the secondary band, while the relapse infection group showed a more moderate increase overall and a minor secondary band (Fig. 5d, e). In combination, the RNA- and protein-based data provide evidence that cure of *T. cruzi* infection is followed by increased GFAP expression in EGCs and/or proliferation of GFAP⁺ EGCs in the colonic myenteric plexus, which may contribute to the recovery of normal transit.

### Analysis of myenteric cellular proliferation after treatment of T. cruzi infection

Our next aim was to investigate the cellular basis of ENS regeneration in our DCD model. We performed pulse-chase EdU labelling experiments to identify cells that were proliferating in the weeks following the end of BZ treatment, one with a short follow-up at 6 weeks post-treatment (wpt) and one with a long follow-up at 27 wpt (Supplementary Fig. 8a). We classified EdU⁺ cells (i.e. progeny of cells that were undergoing DNA replication during at least one of the EdU pulses) into GFAP⁺ (glial) and Hu⁺ (neuronal) co-localising subsets inside and around myenteric plexus ganglia, as well as subsets expressing neither marker (Fig. 6, Supplementary Fig. 8). Of 1105 EdU⁺ cells observed, 246 (22.3%) were intra-ganglionic and the majority of these (191, 77.6%) expressed neither GFAP nor Hu. This subset was significantly more frequent in BZ-treated mice than in uninfected controls (Fig. 6c), consistent with a proliferative tissue repair process in the ENS. The rarity of EdU⁺ cells co-localising with GFAP or Hu expression made a comparison of frequencies tentative. Nevertheless, EdU⁺ GFAP⁺ observations were more frequent in BZ-treated mice, close to statistical significance for the intra-ganglionic site and significantly increased in the peri-ganglionic area in the long follow-up experiment (Fig. 6c, d). Only nine instances of Hu expression co-localising with EdU were observed, all of which were in BZ-treated mice (Fig. 6, Supplementary Fig. 8). Of note, in none of these events did Hu expression match the typical morphology seen for neurons in healthy control mice (Fig. 6b, Supplementary Fig. 8). Thus, while we found evidence of generalised cellular proliferation in the ENS repair phase after anti-parasitic treatment, the frequency and arrangement of co-localising EdU and Hu signals did not provide compelling evidence for proliferative neurogenesis as an explanation for the observed robust recovery of neuron density (Fig. 3).

### Delayed treatment improves GI function to a lesser extent than early treatment

Most cases of human Chagas disease are associated with chronic *T. cruzi* infections. We therefore investigated treatment initiated at 24 wpi, to assess the impact on DCD in the chronic stage (Supplementary Fig. 9a). The bioluminescence profile of the untreated infected mice followed a similar pattern as previously shown (Fig. 7a, b). Treatment with BZ at 24 wpi resulted in elimination of parasite bioluminescence by 30 wpi (Fig. 7a, b), a gradual gain of body weight (Supplementary Fig. 9b) and reversal of splenomegaly (Fig. 7c). Relapses of infection were detected in 30% (3/10) of the treated mice, with reappearance of the bioluminescence signal mostly in the abdominal area (Fig. 7a, b, d). The proportions of cures and relapses were not significantly different from those previously observed for treatment at 6 wpi (Fisher's exact test $P = 0.461$). Ex vivo imaging at end-point necropsy of untreated infected mice (48 wpi) showed the highest intensity and frequency of infection was in the heart and GI tract (Fig. 7e, f, Supplementary Fig. 9g, Supplementary Data 1).

As expected, there was a significantly longer GI transit time ($\bar{x} = 184$ min) in untreated infected mice at all chronic time points compared to uninfected controls (Fig. 8a, b). The small number of relapse cases ($n = 3$) limited our ability to infer the consequences of treatment failure in terms of disease development. Nevertheless, it is noteworthy that the distribution of infections amongst organs and tissues appeared to have a similar profile to what was observed in the acute stage treatment experiment (Fig. 7e, f Supplementary Fig. 9g, Fig. 1, Supplementary Data 1). The relapse mice initially showed strong improvements in transit time post-treatment, but by the end-point, they transitioned towards an increased transit time ($\bar{x} = 174$ min), close to the delay seen in untreated infected mice (Fig. 8a, b). There was no significant alleviation of faecal retention in mice where infection relapsed after treatment (Fig. 8c, Supplementary Fig. 9f). Retrospective comparison of pre-treatment body weights, parasite loads

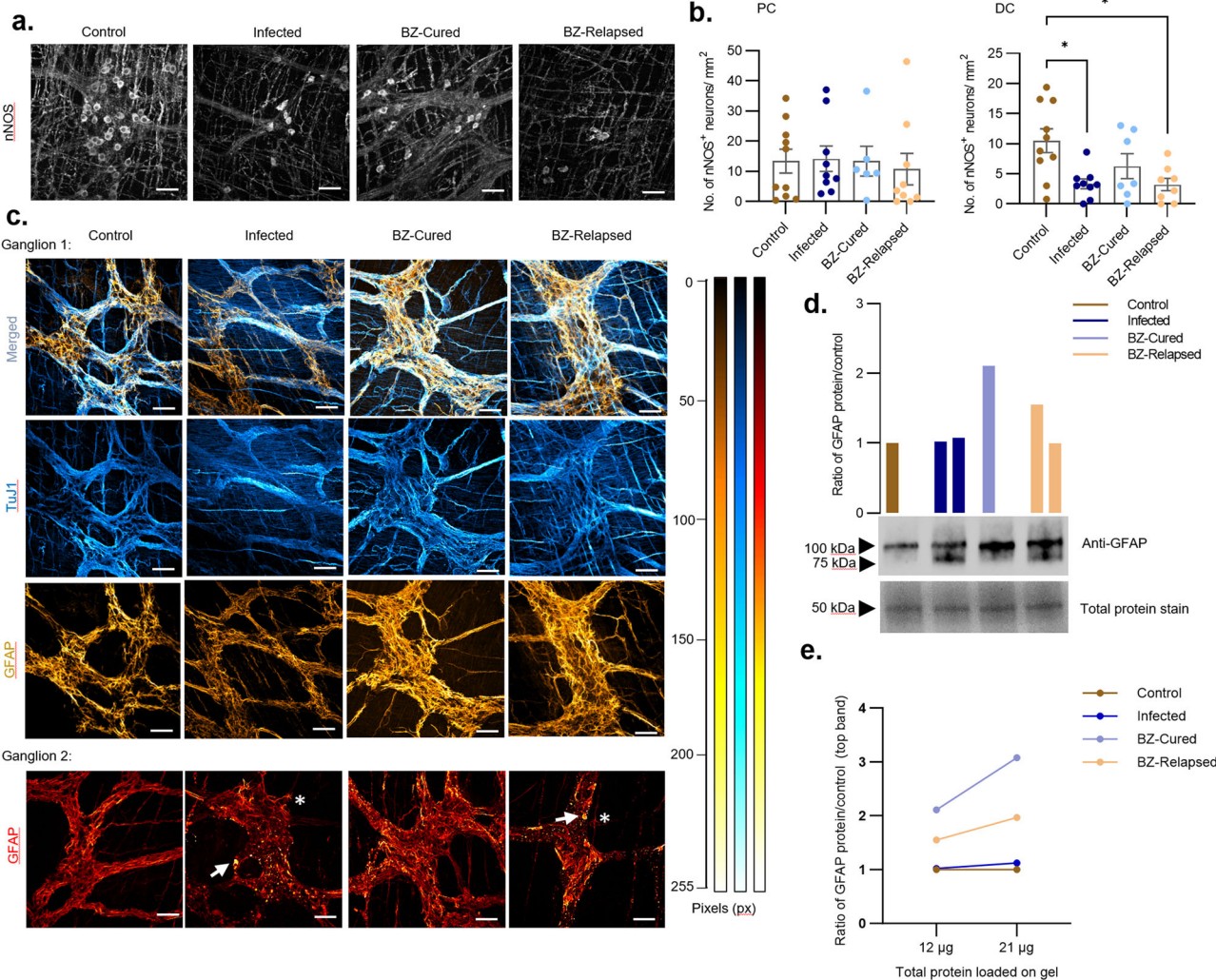

**Fig. 5 | Enteric nitrergic neuron and glial cell dynamics in chronic *T. cruzi* infections and after benznidazole treatment. a** Representative immuno-fluorescent confocal z-stack whole-mount images of nNOS+ neurons in the myenteric plexus of the mouse colon. **b** Bar plots show number of nNOS+ neuronal cell bodies per field of view in control (*n* = 10), infected (*n* = 9), benznidazole treated and cured (BZ-Cured, *n* = 6 in proximal [PC] and *n* = 7 in distal colon [DC]), and benznidazole treated and relapsed (BZ-Relapsed, *n* = 9 in PC and *n* = 8 in DC) C3H/HeN mouse proximal and distal colon regions. **c** Representative immuno-fluorescent confocal z-stack images to display changes in anti-GFAP (gold yellow colour intensity scale) stained enteric glial cells (EGCs) co-labelled with anti-TuJ1 (blue colour intensity scale) enteric neural network across different experimental groups in the colon myenteric plexus. Top panel shows merged images of GFAP and TuJ1 labelled cells. Bottom panel shows images of morphologically diverse GFAP+ EGCs (red pixel colour intensity scale) in the myenteric plexus of infected and BZ-Relapsed colons compared to control and BZ-Cured. White arrows show indicate putative degraded GFAP+ EGC and white stars indicate activated GFAP+ EGC morphologies. All confocal images (**a**, **c**) were taken at 400× magnification, scale bar = 50 μm. Colour heat map scales show pixel intensity. All micrographs are representative images of two independent experiments. **d** Bar plots and **e** paired dot plot show Western blot analysis of GFAP protein abundance in whole tissue lysates from mouse colons (as a ratio of control levels) from infected, BZ-Cured and BZ-Relapsed (*n* = 3, biological samples, all groups). Representative immunoblot in **d** shows α-GFAP staining using 12 μg of lysate (corresponding to the bar plot groups above). To demonstrate equal sample loading, the most abundant protein in each group is presented below as a stain-free gel image. For comparison, GFAP abundance quantified by Western blotting of 12 and 21 μg of each lysate is indicated in plot (**e**).

and transit times showed that there were no significant differences between BZ-treated mice that were cured compared to relapsed (Supplementary Fig. 10).

Transit time in cured mice improved to an intermediate level (x̄ = 144 min), which was stable for the duration of the experiment, indicating a partial recovery of function (Figs. 8a, b). When we analysed colonic faecal retention there was stronger evidence for recovery of GI function, with this constipation phenotype significantly alleviated after BZ-mediated cure of infection (Fig. 8c and Supplementary Fig. 9f). Significant denervation of the ENS was again evident in untreated infections, with 77% loss of Hu+ soma from the colonic myenteric plexus compared to uninfected controls (Fig. 8d, e). Benznidazole-cured mice had qualitatively more typical soma morphologies (Fig. 8d)

and higher average neuron density than these untreated infected mice, but at 57% of normal levels, there remained a significant deficit (Fig. 8e).

In summary, when treatment was delayed until the chronic phase of infection there was more modest recovery of GI transit function and limited evidence of myenteric plexus neuron replenishment. Therefore, the timing of anti-parasitic treatment is likely to be an important factor affecting the degree to which GI function can be restored in DCD.

## Discussion

We investigated the impact of BZ, the front-line treatment for *T. cruzi* infection, in a mouse model that exhibits delayed GI transit and colon

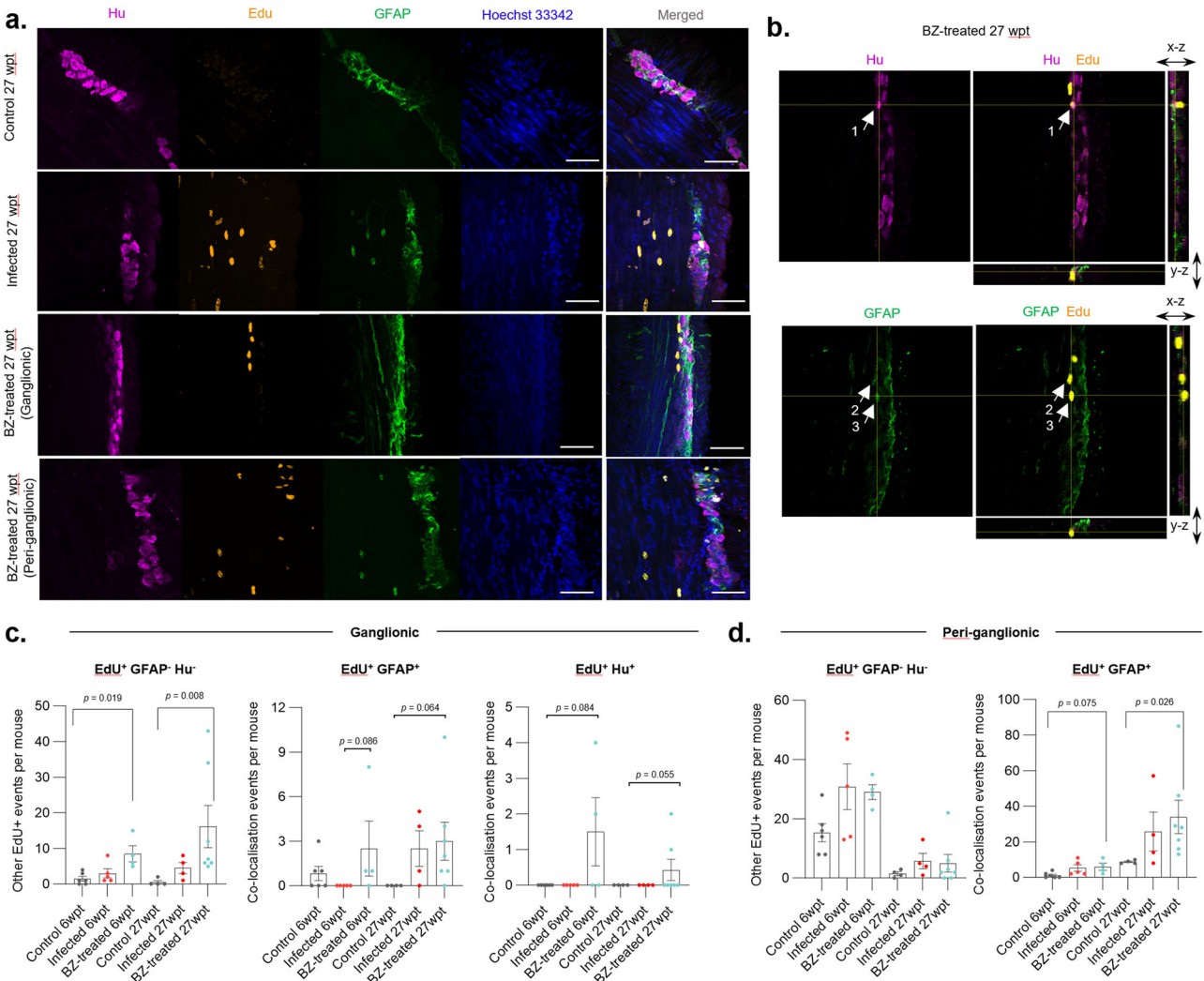

**Fig. 6 | ENS cellular proliferation dynamics after benznidazole treatment of *T. cruzi* infections. a** Representative compressed z-stack images of transverse colon sections at 36 weeks post-infection with *T. cruzi*, 27 weeks post-treatment (wpt) with benznidazole (BZ), 17 weeks after EdU pulse phase. Images shown for female C3H/HeN mice that were uninfected (Control), infected with TcI-JR and (i) vehicle-administered (Infected), or treated with BZ for 20 days at 6 wpi (BZ-treated). All mice were pulsed with 6 doses of EdU between 1 and 7 weeks after withdrawal of BZ to label the progeny of cells that were proliferating during an EdU pulse (orange). Immunofluorescently-labelled Hu[+] neuronal cell bodies (magenta) and GFAP[+] glial cells (green). Hoechst 33342 stain shows DNA (blue). **b** Single z-slice merged image from a BZ-treated mouse showing intra-ganglionic EdU co-localisation events with Hu or GFAP expression (white arrows; cell 1: co-localisation with Hu at z-slice 7/18; cell 2 and 3: co-localisation with GFAP at z-slice 13/18). Orthogonal view of z-planes of the image across the x and y axes are marked by yellow lines. **c, d** Frequencies of EdU[+] cells co-localising (or not) with Hu or GFAP protein expression in myenteric ganglionic (**c**) and peri-ganglionic (**d**) locations (6wpt: control *n* = 6, infected *n* = 5 and BZ-Cured *n* = 4; 27 wpt: control *n* = 4, infected *n* = 4 and BZ-Cured *n* = 7). Data also include a shorter follow-up cohort (12 weeks post-infection, 6 weeks post-treatment (wpt), 1 week after the end of the EdU pulse phase). Data are expressed as mean ± SEM, data points are for individual mice. Statistical significance was tested using Kruskal–Wallis tests (*P*-values < 0.1 are annotated). All confocal images were taken at 400× magnification, scale bar = 50 μm.

myenteric plexus denervation, key features of DCD. When a parasitological cure was achieved, this halted disease progression and reversed symptoms associated with GI transit delay. This recovery was associated with partial, yet significant, restoration of myenteric neuron density in the colon. Gene expression profiling and analysis of cellular proliferation post-treatment showed that functional cure and ENS regeneration was associated with resolution of chronic inflammation and a switch to a proliferative repair response. Furthermore, our results emphasise the importance of the timing of treatment initiation, showing that intervention at 6 weeks had a greater impact than at 24 weeks post-infection.

Treatment for DCD is limited to palliative dietary adjustments and surgical interventions with significant mortality risk[16]. Lack of data has prevented a consensus on whether *T. cruzi*-infected adults who are asymptomatic should be treated with anti-parasitic chemotherapy[4,12,40]. Our findings in a pre-clinical mouse model provide evidence that prompt treatment with BZ can prevent chronic DCD. When treatment was delayed until the chronic phase, sterile cure of infection only resulted in a partial GI functional recovery and less evidence of ENS regeneration, indicating that some tissue pathology reaches an irreversible stage. This echoes the results of a clinical trial of BZ in late-stage chronic Chagas cardiomyopathy patients, in which the drug performed no better than a placebo in preventing disease progression or death[8]. Further work is required to determine the point at which the cure of infection will cease to yield functional improvement in DCD. Nonetheless, our findings provide a pre-clinical in vivo evidence base supporting the concept that the earlier anti-parasitic chemotherapy is initiated, the greater the chances of preventing or delaying the progression of digestive disease.

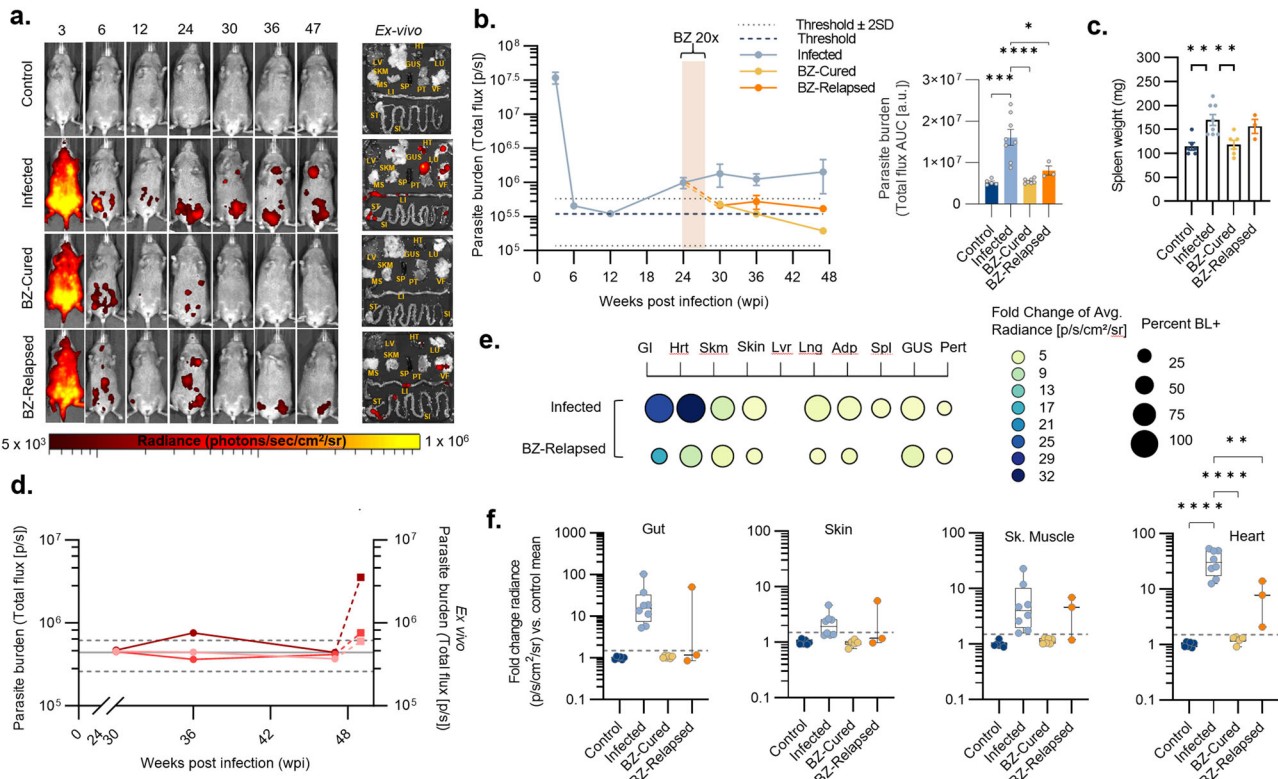

**Fig. 7 | Impact of chronic phase benznidazole treatment in digestive Chagas disease mice. a** Representative in vivo bioluminescence (BL) images of female C3H/HeN mice that were uninfected (Control), infected with TcI-JR and (i) vehicle-administered (Infected), or treated with benznidazole at 24 weeks post-infection (wpi) and (ii) assessed as parasitologically cured (BZ-Cured), or (iii) assessed as treatment failure (BZ-Relapsed). Ex vivo images show bioluminescence in liver (LV), lungs (LU), gut mesenteries (MS), heart (HT), spleen (SP), skeletal muscle (SKM), visceral fat (VF), stomach (ST), small intestine (SI), large intestine (LI), genito-urinary system (GUS), and peritoneum (PT). BL intensity expressed using radiance (p/s/cm²/sr) pseudocolour heat maps. **b** In vivo BL profiles of infected (*n* = 8, except *n* = 19 at 6 and 12 wpi, *n* = 16 at 24 wpi and *n* = 20 at 3 wpi), BZ-Cured (*n* = 7) and BZ-Relapsed (*n* = 3) mice over time. Dashed lines show uninfected control autoluminescence-based thresholds. Bar plots show cumulative parasite burdens based on area under the curve of the line plots (control *n* = 5, infected *n* = 8, BZ-Cured *n* = 7 and BZ-Relapsed *n* = 3). **c** Spleen weights of control (*n* = 6), infected (*n* = 8), BZ- Cured (*n* = 6) and BZ-Relapsed (*n* = 3) groups. **d** Post-treatment in vivo BL infection profiles for individual BZ-Relapsed mice; thresholds as in (**b**). **e** Mean tissue-specific infection intensities in untreated and relapsed infections. Circle size indicates the percentage of individual animals with BL-positive (BL+) signal for each sample type; colour indicates infection intensity (fold change in ex vivo BL vs uninfected controls). **f** Tissue-specific infection intensities for gut, heart, skeletal muscle and skin of control (*n* = 6) infected (*n* = 8), BZ-Cured (*n* = 6) and BZ-Relapsed (*n* = 3) mice. Data expressed as mean fold change in radiance vs. uninfected mean. Threshold line is the mean for an internal control (empty) region of interest. Box plots show the median with minimum and maximum values as whiskers; the bounds of the box show IQR. All other data are expressed as mean ± SEM. Statistical significance was tested using a two-sided *t*-test or one-way ANOVA with Tukey's HSD test. Only significant differences are annotated: *P < 0.05, **P < 0.01, ***P < 0.001, **** *P < 0.0001.

In the context of DCD, our results provide insight into the dynamic relationships between *T. cruzi* infection, host responses, ENS damage and tissue repair. Most denervation occurred in the acute phase of infection, yet the complete and sustained normalisation of GI transit function after BZ treatment at 6 weeks shows that these acute losses are not sufficient to explain chronic disease symptoms, contrary to early theories of DCD aetiology[18]. Transient functional improvement in untreated control infections in the early chronic phase, between 6 and 12 weeks, further supports this conclusion (Fig. 2a[37]). Over time, chronic infection of the GI tract led to further neuron losses and a gradual decline in GI function. Moreover, in cases of failed treatment, relapses of *T. cruzi* infection were associated with a return to GI dysfunction. Together, these data support the conclusion that chronic infection actively drives disease, as has been circumstantially suspected from the detection of parasites in GI tissues from human DCD patients[19–27]. However, the disease aetiology is more complex than anticipated because neither parasite load, nor the degree of denervation directly predicted the severity of functional impairment in individual animals (Supplementary Fig. 2e)[37]. Also, the temporary improvement in transit time at 12 weeks in untreated infections was not associated with recovery of myenteric neuron density. This might

be explained by compensatory plasticity of the remaining ENS and/or extrinsic circuitry in the denervated colon[41]. As seen in the CNS, it is also possible that the existing ENS neural circuit rewires to compensate for the neuronal loss by rebalancing the excitatory and inhibitory outputs of the network[42]. Our nervous and immune system gene expression analysis indicated that the broader balance of inflammatory, regulatory and tissue repair factors at play in the infected colon also contributes to the DCD phenotype. For example, nerve growth factor was one of only five genes that were significantly upregulated in all the experimental groups compared to uninfected controls, and the only one at *P* < 0.01. Some upregulated genes with ENS functions were shared amongst cures and relapses e.g. *Nefl*, *Plp1*, *Nos1*, whilst others were unique to the cures e.g. *Gfap*, *Rbfox3* (NeuN) (Supplementary Data 2). Thus, some tissue repair processes may co-occur with chronic inflammation during active infections, likely in distinct microdomains of the colon, but the full engagement of a regenerative ENS repair programme appears to be dependent on complete clearance of the infection.

While the TcI-JR-C3H/HeN model is characterised by widely disseminated chronic infections, in several others (e.g. BALB/c, C57BL/6) *T. cruzi* is mainly restricted to the stomach, colon and skin[29,43]. Why the

GI tract serves as a long-term permissive site for *T. cruzi*, at least in mice, has been unclear because there have been minimal data on gut-specific immune responses. At the transcriptional level, we found that chronic infection of the colon is associated with a robust type 1

inflammatory response, dominated by markers of CD8[+] T cell recruitment, similar to that seen in studies of other tissue types[10,44–48]. The discovery that at least 9 genes with immuno-inhibitory potential were also upregulated suggests that there are host-intrinsic mechanisms

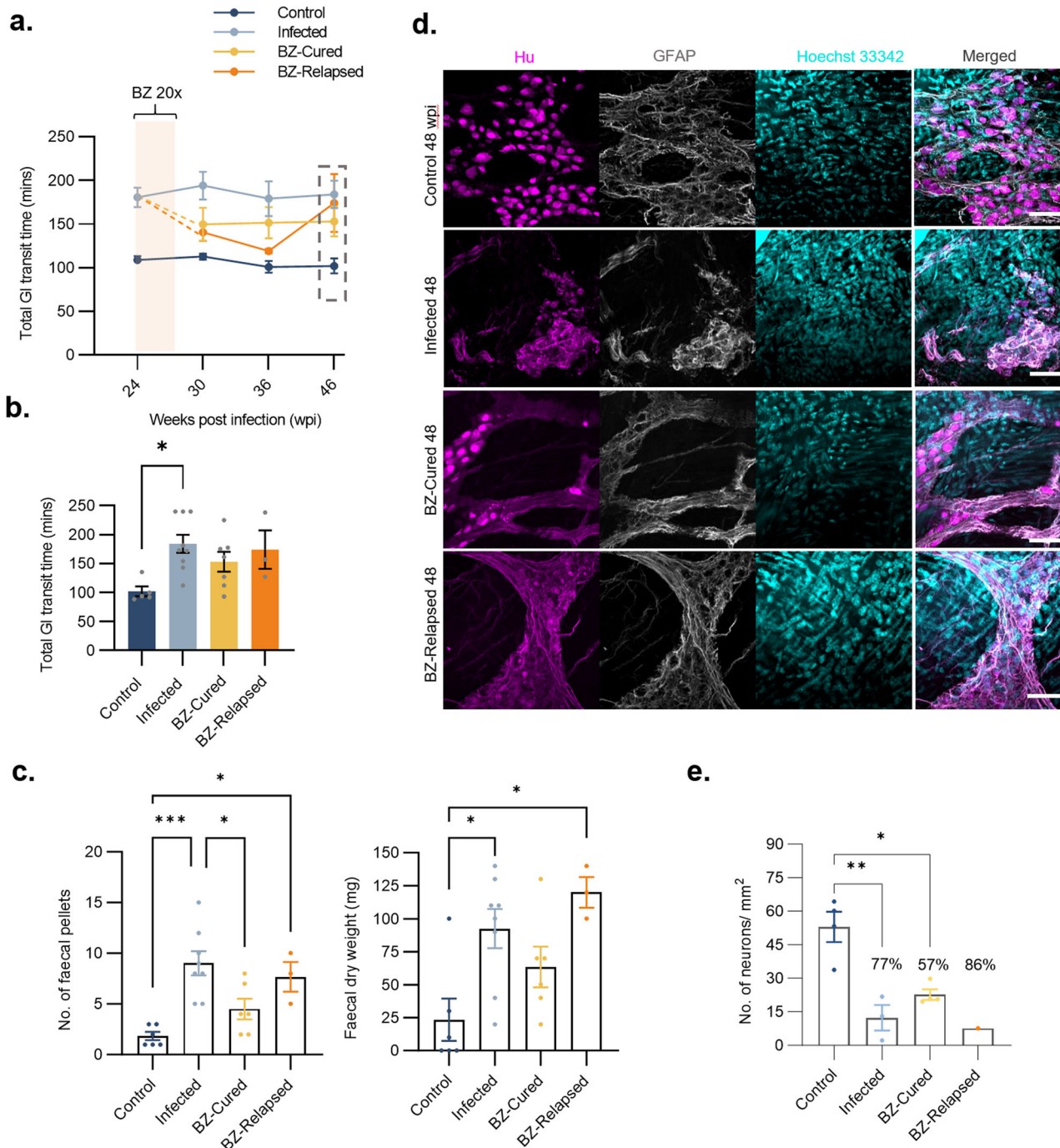

**Fig. 8 | Impact of chronic phase benznidazole treatment on the ENS and GI transit function. a** Total GI transit times of female C3H/HeN mice that were uninfected (Control; $n = 5$, except $n = 11$ at 24 wpi and $n = 6$ at 30 wpi), infected with TcI-JR and (i) vehicle-administered (Infected; $n = 9$, except $n = 23$ at 24 wpi and $n = 8$ at 30 wpi), or treated with benznidazole at 24 wpi and (ii) assessed as parasitologically cured (BZ-Cured; $n = 7$), or (iii) assessed as treatment failure (BZ-Relapsed; $n = 3$). Cream bar shows the BZ treatment window (24–29 wpi). **b** Bar plot shows total GI transit times of control ($n = 5$), infected ($n = 9$), BZ-Cured ($n = 7$) and BZ-Relapsed ($n = 3$) at the 48 wpi end-point. Control vs Infected $P = 0.015$. **c** Faecal pellet analyses show the number of faecal pellets and dry faecal weight of control ($n = 6$), infected ($n = 8$), BZ-Cured ($n = 6$) and BZ-Relapsed ($n = 3$) mice after 4 h'

fasting. For pellet counts, Control vs Infected $P = 0.0003$; Control vs BZ-Relapsed $P = 0.024$; Infected vs BZ-Cured $P = 0.023$. For faecal weight, Control vs Infected $P = 0.017$; Control vs BZ-Relapsed $P = 0.001$. **d** Representative compressed z-stack whole-mount immunofluorescence images of myenteric Hu[+] neurons and GFAP[+] glial cells in colon samples at 48 wpi. **e** Bar plots show the density of Hu[+] neurons in proximal colon myenteric plexus of control ($n = 4$), infected ($n = 3$), BZ-Cured ($n = 4$) and BZ-Relapsed ($n = 1$) groups. Control vs Infected $P = 0.003$; Control vs BZ-Cured $P = 0.012$. Data are expressed as mean ± SEM. Statistical significance was tested using one-way ANOVA followed by Tukey's HSD test. Only significant differences are annotated: *$P < 0.05$, **$P < 0.01$, ***$P < 0.001$. All immunofluorescence images were taken at 400× magnification, scale bar = 50 μm.

that limit tissue damage, yet also enable *T. cruzi* persistence. We previously found that chronic *T. cruzi* infection is restricted to a few rare foci, which can be widely scattered within the colonic smooth muscle layers and are regularly re-seeded in new locations by motile trypomastigotes[37,43]. Here, we analysed RNA from full-thickness colon tissue and, for selected markers of neurons and glia, immunofluorescence analysis of the myenteric plexus. Therefore, a key challenge now will be to determine how infection and immune response dynamics relate to specific cell types, and how these in turn shape pathology at finer spatial and temporal scales.

The extent to which the adult ENS is capable of repair and regeneration is a fundamental question, with relevance across diverse enteric neuropathies. We observed a recovery of myenteric plexus neuron numbers in mice several months after the sterile cure of *T. cruzi* infection. There are several potential neurogenic mechanisms that could underlly this replenishment. It has been proposed that enteric neurons are regularly replaced from a population of neural stem cells as part of gut homoeostasis[49], however, the majority of studies indicate that enteric neurogenesis is absent, or extremely limited in the steady-state adult gut[50–55]. Regeneration of the ENS involves neurogenesis from EGC pre-cursors after chemical injury using benzalkonium chloride (BAC) or Dextran Sulfate Sodium Salt (DSS)[51,52,54,55] and from extrinsic Schwann cell pre-cursors in mouse models of Hirschprung disease[56,57]. Our approach did not enable us to define the ontogeny of new neurons, but we did investigate whether cellular proliferation might be involved in ENS regeneration. Using EdU incorporation assays, we identified abundant progeny of cells that were replicating in the weeks following BZ treatment later residing within and around myenteric ganglia. Only in rare cases (<1%) did EdU co-localise with the neuronal marker Hu and there were no convincing examples of typical soma morphologies with EdU+ nuclei. This appears incongruent with the hypothesis that proliferative neurogenesis underlies the observed re-innervation after cure of *T. cruzi* infection. However, our EdU exposure windows may have been insufficiently broad to capture many replicating neuronal pre-cursors - this molecule has an in vivo half-life of only 25 min[58]. Alternatively, neurogenesis may more often occur through transdifferentiation without proliferation. Given recent work on the neurogenic potential of EGCs[52,59], it is notable that we saw upregulation of the canonical glial markers GFAP and S100β specifically in cured mice, as well as PLP1, which corresponds to the subset of glial cells that differentiate into neurons in the DSS colitis model[52]. Transcriptional changes in healing colon tissue also comprised upregulation of the stem cell transcription factor *Zeb1*, which regulates epithelial–mesenchymal transition[60] and neuronal differentiation in the CNS[61], including from radial glial-like cells in the adult hippocampus[62]. We also saw upregulation of *Ngf* and *Notch2*, critical regulators of neurogenesis; Notch signalling, in particular, has been implicated in the maintenance of the neural progenitor pool and controlling glial to neuronal differentiation in the CNS[63] and the gut[64,65]. The *T. cruzi* infection and cure approach may therefore open new experimental opportunities to study adult neurogenesis and inform the development of regenerative therapies for enteric neuropathies, most pertinently DCD.

## Methods

### Ethics statement
Research was conducted in compliance with all relevant legal and ethical regulations. All animal procedures were performed under UK Home Office project license P9AEE04E, approved by the Animal Welfare and Ethical Review Board of the London School of Hygiene and Tropical Medicine, and in accordance with the UK Animals (Scientific Procedures) Act (ASPA) 1986.

### Parasites
This study used the TcI-JR strain of *T. cruzi* parasites constitutively expressing the red-shifted firefly luciferase variant *PPy*RE9h[29].

Epimastigotes were cultured in vitro in a supplemented RPMI-1640 medium at 28 °C under selective drug pressure with 150 μg ml⁻¹ G418. MA104 monkey kidney epithelial cell monolayers were infected with metacyclic trypomastigotes, obtained from stationary phase cultures, in MEM media + 5% FBS at 37 °C and 5% $CO_2$. After 5–10 days, tissue culture parasites (TCTs) were harvested from the culture supernatant and aliquots from a single batch were cryopreserved in 10% DMSO. For in vivo infections, TCTs were thawed at room temperature, sedimented by centrifugation at $10,000 \times g$ for 5 min, washed in 1 ml complete medium, sedimented again and resuspended in 250 μl complete medium. After 1 h incubation at 37 °C, active parasites were counted and the suspension was adjusted to the required density.

### Animals and infections
Female C3H/HeN mice, aged 6–8 weeks, were purchased from Charles River (UK). Female CB17 SCID mice, aged 8–12 weeks were bred in-house or purchased from Charles River (UK). Mice were housed in individually ventilated cages on a 12 h light/dark cycle and habituated for 1–2 weeks before experiments. They had access to food and water available *ad libitum* unless otherwise stated. The vivarium was maintained at 22 °C and 45% humidity. Mice were maintained under specific pathogen-free conditions. Humane end-points were loss of >20% body weight, reluctance to feed or drink freely for more than 4–6 h, and loss of balance or immobility.

Inocula of $1 \times 10^5$ TCTs were used to infect SCID mice via i.p. injection. After 3 weeks, motile blood trypomastigotes (BTs) were derived from the supernatant of cardiac whole blood after passive sedimentation of mouse cells for 1 h at 37 °C. C3H/HeN mice were infected with $1 \times 10^3$ BTs in 0.2 ml PBS via i.p. injection. The BZ treatment schedule was 100 mg kg⁻¹ day⁻¹ for 20 consecutive days via oral gavage. Benznidazole was prepared from powder form at 10 mg ml⁻¹ by dissolving in vehicle solution (0.5% w/v hydroxypropyl methylcellulose, 0.5% v/v benzyl alcohol, 0.4% v/v Tween 80 in deionised water).

At experimental end-points, mice were sacrificed by ex-sanguination under terminal anaesthesia (Euthatal/Dolethal 60 mg kg⁻¹, i.p.) or by cervical dislocation. Selected organs and tissue samples were cleaned with PBS and snap-frozen on dry ice, fixed in 10% Glyofixx or transferred to ice-cold DMEM medium to suit different downstream analysis methods.

### Total GI transit time assay
Carmine red dye solution, 6% (w/v) in 0.5% methylcellulose (w/v) in distilled water was administered to mice by oral gavage (200 μl). Mice were returned to their home cage for 75 min, after which they were placed in individual containers and observed. The time of excretion of the first red-stained faecal pellet was recorded and the mouse was returned to its cage. A cut-off time of 4 h was employed as the maximum GI transit delay for the assay for welfare reasons. Total GI transit time was calculated as the time taken to expel the first red pellet from the time of gavage.

### In vivo bioluminescence imaging
Mice were injected with 150 mg kg⁻¹ D-luciferin i.p., then anaesthetised using 2.5% (v/v) gaseous isoflurane in oxygen. Bioluminescence imaging was performed after 10-20 min using an IVIS Lumina II or Spectrum system (PerkinElmer). Image acquisition settings were adjusted depending on signal saturation (exposure time: 1–5 min; binning: medium to large). After imaging, mice were placed on a heat pad for revival and returned to cages. Whole body regions of interest (ROIs) were drawn on acquired images to quantify bioluminescence, expressed as total flux (photons/second), to estimate in vivo parasite burden in live mice[28]. The detection threshold was determined using uninfected control mice. All bioluminescence data were collected and analysed using Living Image v4.7.3.

## Ex vivo bioluminescence imaging

Food was withdrawn from cages 4 h prior to euthanasia. Five to seven min prior to euthanasia, mice were injected with 150 mg kg$^{-1}$ D-luciferin i.p. After euthanasia, mice were perfused transcardially with 10 ml of 0.3 mg ml$^{-1}$ D-luciferin in PBS. Typically, organs collected included the heart, liver, spleen, lungs, skin, peritoneum, the GI tract, the genitourinary system and their associated mesenteries, caudal lymph nodes, as well as samples of hindlimb skeletal muscle and visceral adipose. These were soaked in PBS containing 0.3 mg ml$^{-1}$ D-luciferin prior to imaging, which was performed as described above. Parasite load in each organ or tissue sample was quantified as a measure of infection intensity. To do this, bioluminescence per organ/tissue was calculated by outlining ROIs on each sample and expressed as radiance (photons s$^{-1}$ cm$^{-2}$ sr$^{-1}$). Radiances from equivalent organs/tissues of age-matched, uninfected control mice were measured and the fold change in bioluminescence for each test organ/tissue was calculated.

## Treatment outcomes

Each individual animal that had been infected and treated with BZ was assessed as either parasitologically cured or relapsed. The limits of detection for in vivo and ex vivo bioluminescence imaging are estimated to be approximately 100 parasites and 20 parasites respectively[28,43]. Mice in which *T. cruzi* was detected on any of the following criteria were assessed as relapses: (i) an in vivo total flux signal $>6.17 \times 10^5$ p/s (uninfected mean +2 SD) in any of the post-treatment imaging sessions; (ii) one or more organ or tissue samples with an ex vivo fold change in radiance greater than the highest measurement obtained across all samples from the uninfected control group ($n = 254$ from 19 mice for 6 weeks treatment and $n = 78$ from 6 mice for 24 weeks treatment). The ex vivo images for all remaining BZ-treated animals were then manually inspected for (iii) the presence of any discrete bioluminescence foci for which the signal radiated from a point source as the threshold was gradually reduced to noise (characteristic of an infection focus). This was necessary to distinguish them from sporadic, weak auto-luminescence signals that co-localised with food and faeces in the GI tract lumen. Independent manual inspection calls were made by two investigators and then cross-checked. For the purposes of this study, a sustained failure to detect *T. cruzi* using these methods was interpreted as a sterile cure.

## Faecal analyses

Isolated colon tissue was cleaned externally with PBS and faecal pellets were gently teased out of the lumen. Faecal pellets were counted and collected in 1.5 ml tubes. Wet weights were recorded and then the tubes were left to dry in a laminar flow cabinet overnight; dry weights were measured the following day.

## Histopathology

Paraffin-embedded, fixed tissue blocks were prepared and 3–5 μm sections were stained with haematoxylin and eosin[29]. Images were acquired using a Leica DFC295 camera attached to a Leica DM3000 microscope. For analysis of inflammation, nuclei were counted automatically using the Leica Application Suite v4.5 software (Leica).

## Immunofluorescence analysis

After necropsy, excised colon tissues were transferred from ice-cold DMEM to PBS. For full-thickness transverse sections of colon tissue, 1 cm colon pieces were fixed overnight in paraformaldehyde (4% w/v in PBS), washed in 15% sucrose solution (15% D-sucrose w/v in PBS) and stored in 30% sucrose solution (30% D-sucrose w/v, 0.01% NaN$_3$ w/v in PBS) at 4 °C. For cryosectioning, tissues were washed in PBS for 5 min at room temperature and then embedded in sucrose and gelatine in PBS (10% w/v, 7.5% w/v), using peel-away moulds (Merck). Blocks were cooled for 45 min at 4 °C then rapidly frozen in an isopentane and dry ice slurry for 1 min. Blocks were mounted in a

cryostat (Leica CM1950) using OCT and 50 μm transverse sections were collected onto Superfrost PLUS slides (Epredia). For colonic muscularis whole-mount preparations, tissues were cut open along the mesentery line, rinsed with PBS, and then stretched and pinned on Sylgard 184 plates. Under a dissection microscope, the mucosal layer was carefully peeled away using forceps and the remaining muscularis wall tissue was fixed in paraformaldehyde (4% w/v in PBS) for 45 min at room temperature.

Fixed samples were washed with PBS for 45 min, with 3 changes, at room temperature and permeabilised with PBS containing 0.5% Triton X-100 for 2 h, followed by blocking for 1 h (10% sheep serum in PBS containing 0.5% Triton X-100). Tissues were incubated with combinations of the following primary antibodies in PBS containing 0.5% Triton X-100 for 48-96 h at 4 °C: mouse anti-HuC/D IgG clone 16A11 at 1:200 (ThermoFisher), rabbit anti-tubulin β−3 polyclonal IgG at 1:500 (Biolegend), rat anti-GFAP monoclonal IgG clone 2.2B10 at 1:500 (ThermoFisher), rabbit anti-nNos polyclonal IgG at 1:500 (ThermoFisher), rabbit anti-cleaved caspase-3 (Asp175) monoclonal IgG clone 269518 (R&D Systems) at 1:250, human anti-Hu sera ("ANNA-1") at 1:25,000. Tissues were washed with PBS for 30 min with three changes, then incubated with appropriate secondary IgG combinations: goat anti-mouse AF546, goat anti-rabbit AF633, goat anti-rat AF546, goat anti-human AF647, donkey anti-rabbit AF488, all at 1:500 (ThermoFisher) in PBS containing 0.5% Triton X-100 for 2 h. DNA was stained with Hoechst 33342 (1 μg ml$^{-1}$) or DAPI (1.5 μg ml$^{-1}$) at room temperature. Tissues used to detect apoptosis were incubated in a TUNEL reaction mixture for 1 h prior to immuno-labelling as per the manufacturer's protocol (In Situ Cell Death Detection Kit, Roche). Control tissues were incubated with only secondary antibodies (without primary antibodies) to assess antibody specificity. Tissues were mounted on glass slides using FluorSave mounting medium (Merck).

Whole mounts were examined and imaged with an LSM880 confocal microscope using a 40× objective (Zeiss, Germany). Images were captured as z-stack scans of 21 digital slices with interval of 1 μm optical thickness. Five z-stacks were acquired per region (proximal and distal colon), per animal. Cell counts were performed on z-stacks after compression into a composite image using the cell counter plug-in of FIJI software[66]. Neuronal density was calculated as the number of HuC/D$^+$, ANNA-1+ or nNOS$^+$ neuron cell bodies (soma) per field of view. HuC/D signal was associated with high background outside ganglia in samples from infected mice, attributed to binding of the secondary anti-mouse IgG to endogenous IgG, so ENS-specific analysis was aided by anti-TuJ1 co-labelling and assessment of soma morphology.

## In vivo 5-ethynyl-deoxyuridine (EdU) labelling

The thymidine analogue 5-ethynyl-deoxyuridine (EdU) stock was administered via 0.2 ml i.p. injections at 25 mg kg$^{-1}$ in 6 doses over several weeks (Supplementary Fig. 8a). Incorporation of EdU was detected using Click-iT Plus EdU Imaging kits (Invitrogen) either 1 or 17 weeks after the final EdU dose. Briefly, PFA-fixed frozen colon transverse sections or whole-mount colon muscularis samples were permeabilised with PBST (1% Triton X-100 in PBS) for 45 min, washed in PBS for 5 min and stained with the EdU labelling solution for 30 min. Samples were then washed in PBS for 5 min prior to entry at the primary antibody labelling stage of the immunofluorescence workflow described above.

Images were captured as z-stack volumes comprising 12 digital slices of 2 μm (frozen sections), with 10 volumes acquired per mouse at 400× magnification. For the 1 week follow-up cohort only, z-stack volumes comprising 15 digital slices of 1 μm centred on the myenteric plexus (whole mounts) were also acquired with 5 volumes per mouse. EdU$^+$ cell counts and co-localisation with GFAP and Hu protein expression were determined by manual inspection of each z-slice of each volume in ZEN (Zeiss) and/or FIJI[66].

## Contractility

Colonic tissue samples were dissected, weighed and placed in an oxygenated Krebs solution. Tissue strips were suture-mounted in tissue baths (10 ml, Panlab Two Chamber Compact Organ Bath, ML0126/10-220; ADInstruments Ltd, UK) connected to force transducers (MLT0420, ADInstruments Ltd, UK) and bridge amplifiers (FE221, ADInstruments Ltd, UK). Tissues were equilibrated in oxygenated Krebs solution for 60 min at 37 °C and an initial tension of 0.5 g was maintained. During equilibration, the tissues were perfused with three washes of oxygenated Krebs solution. Basal activity was recorded and collected using PowerLab 2/26 data acquisition system (PL2602/P, ADInstruments Ltd, UK), and analysed using LabChartPro v8.

## Western blot

Frozen colon tissue samples were lysed in RIPA Buffer and the total protein concentration was quantified using a BCA assay kit as per manufacturer's protocol (Thermo Scientific). Lysates were obtained from three independent biological samples per group and pooled into a single sample for analysis. Polyacrylamide gel electrophoresis was performed to separate proteins using 4–20% stain-free TGX gels (Bio-Rad). Proteins were visualised by UV-induced fluorescence using a Chemidoc imaging system (Bio-Rad) to verify equal loading of samples. The most abundant protein band in each loading control sample was used for quantification. Proteins were transferred to nitrocellulose membranes in a trans-blot turbo transfer system (Bio-Rad). Membranes were blocked for 30 min using 5% skimmed milk in PBST and then probed with rat anti-GFAP primary antibody (1:2000, cat # 13-0300, ThermoFisher) overnight at 4 °C, followed by incubation with HRP-conjugated goat anti-rat secondary antibody (1:5000, cat # 31460, ThermoFisher) for 2 h at room temperature and visualisation using enhanced chemiluminescence (ECL kit, GE Healthcare Life Sciences). Data were analysed using the gel analysis package in FIJI[66].

## RNA extraction

Frozen tissue samples were thawed in 1 ml Trizol (Invitrogen) per 30–50 mg tissue and immediately homogenised using a Precellys 24 homogeniser (Bertin). 200 μl of chloroform was added to each sample and mixed by vortex. The aqueous phase was separated by centrifugation at $13,000 \times g$ at 4 °C and RNA was purified using the RNeasy Mini Kit (Qiagen) with on-column DNASe digestion, as per manufacturer's protocol. A Qubit Fluorimeter (Thermofisher) and/or a Nanodrop instrument was used to assess RNA quality and quantity.

## RT-qPCR

cDNA was synthesised from 1 μg of total RNA using Superscript IV VILO master mix (Invitrogen), as per manufacturer's protocol, in reaction volumes of 20 μl. qPCR reactions were carried out using QuantiTect SYBR green master mix (Qiagen) with 200 nM of each primer and 4 μl of cDNA diluted 1/50 in DEPC water. Reactions were run using an Applied Biosystems Fast 7500 machine (Thermofisher) as per the manufacturer's protocol.

A final cDNA volume of 100 μl was made by adding RNase-free DEPC water (1:5 dilution) and stored at −20 °C until further use. qPCR reactions consisted of 10 μl QuantiTect SYBR green master mix, 6 μl of forward and reverse primer mix (200 nM; Supplementary Table 1) and 4 μl of cDNA diluted 1/50 in DEPC water. For No-RT and no template control reactions, 4 μl of solution from the No-RT cDNA reaction and DEPC water were added respectively. Reactions were run using an Applied Biosystems Fast 7500 machine (ThermoFisher) as per the manufacturer's protocol. Data were analysed by the $\Delta\Delta C_t$ method[67] using murine *Oaz1* as the endogenous control gene.

## Nanostring gene expression analysis

RNA was adjusted to 30–60 ng μl$^{-1}$ and analysed on a Nanostring nCounter system (Newcastle University, UK). We used a "PanelPlus" set of target probes comprising the standard mouse immunology nCounter codeset (XT-CSO-MIM1-12) and a custom selection of 20 probes from the mouse neuroinflammation and neuropathology codesets: *Acaa1a, Adora2a, Cck, Ch25h, Cidea, Drd1, Drd2, Gfap, MAPt, Nefl, Ngf, Nos1, NPY, P2rx7, Pla2g4a, Pla2g5, Plp1, Rbfox3, S100b* and *Tubb3*. The core immunology codeset comprised 547 protein-coding test genes, 14 housekeeping control genes, 6 positive binding control probes and 8 negative binding control probes. Fifty nine test genes were below a detection threshold limit (mean negative control bound probe count + 3 SDs) and were excluded from the analysis. The final codeset comprised probes for 508 test genes and was analysed using nSolver v4.0. Data were normalised in the Basic Analysis module with positive control and housekeeping gene normalisation probe parameters both set to the geometric mean. Normalised data were then imported to the Advanced Analysis module and used to analyse differential gene expression between groups and pathway scores using default parameters. Samples were annotated with their run number as a confounding variable.

## Sample size and statistics

A power calculation was performed using pilot data for total GI transit time (min) in infected vs uninfected mice ($\bar{x}_1 - \bar{x}_2 = 163 - 98$, mean delay = 65, SD = 35). The primary outcome effect size was ≥70% reversal of transit time delay after BZ-mediated cure of infection (target post-treatment $\bar{x}_3 = 117$ min). Calculations were carried out using the NC3Rs Experiment Design Assistant (https://eda.nc3rs.org.uk/) for power = 0.8 and α = 0.05. The inferred sample size was $n = 10$ per experiment. An additional 5 mice were allocated to the BZ treatment group to account for a predicted 2:1 ratio of cures to relapses. For treatment at 6 wpi, data were pooled from two independent experiments. For treatment at 24 wpi, data are from one experiment. Sample sizes were reduced for some data sets due to attrition of mice associated with progression to a humane end-point before the end of an experiment ($n = 6$) and imaging equipment faults ($n = 1$).

Individual animals were used as the unit of analysis. No blinding or randomisation protocols were used. Statistical differences between groups were evaluated using 2-tailed, unpaired Student's $t$-tests, one-way ANOVA with Tukey's post-hoc correction for multiple comparisons, or Fisher's exact test. These tests were performed in nSolver 4.0, GraphPad Prism v9 or R v3.6.3. Differences of $P < 0.05$ were considered significant.

## Reporting summary

Further information on research design is available in the Nature Portfolio Reporting Summary linked to this article.

## Data availability

Source data are provided with this paper.

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

## Acknowledgements

We thank Hernán Carrasco for sharing parasite strains, Vanda Lennon for the ANNA-1 reagent, Jody Phelan for helping with R scripts and the LSHTM Biological Services Facility staff for technical support and animal husbandry. The work was supported by an MRC grant (MR/T015969/1) awarded to J.M.K., an EU Marie Curie Fellowship (grant agreement no. 625810) awarded to M.D.L., and an MRC New Investigator Research Grant (MR/R021430/1) awarded to M.D.L.

## Author contributions

M.D.L. designed the study and acquired funding with support from M.C.T., C.J.M. and J.M.K. All authors contributed to the design of discrete experiments and analyses. A.A.K., H.C.L., L.W. and M.D.L. conducted the experiments and analysed data with technical support and conceptual advice from R.R., S.J., A.F.F., M.C.T., C.J.M. and J.M.K. A.F.F., C.J.M. and J.M.K. provided reagents. A.A.K. and M.D.L. prepared the manuscript with reviewing and editing input from H.C.L., A.F.F., M.C.T., C.J.M. and J.M.K.

## Competing interests

The authors declare no competing interests.
