## [Peer Review File · Nature Communications]

Enteric nervous system regeneration and functional cure of experimental digestive Chagas disease with trypanocidal chemotherapyREVIEWER COMMENTS

Reviewer #1 (Remarks to the Author):

- What are the noteworthy results?

The manuscript by Kahn et al. describes novel findings in the murine model of *Trypanosoma cruzi* infection that advance our understanding of the pathogenesis of Chagas disease. Specifically, the data relate to the effects of *T. cruzi* infection on the enteric nervous system and “digestive Chagas disease” (DCD). They show that curative treatment with the drug, benznidazole, leads to the recovery of normal GI function if administered early enough (6 weeks post-infection). Whereas later treatment (24 wks post-infection) result in only partial reversal of DCD. The research includes gene expression profiling and immunofluorescence studies to characterize the effects of the treatments on immune and neural pathways.

- Will the work be of significance to the field and related fields? How does it compare to the established literature? If the work is not original, please provide relevant references.

The work has the important implication that early (as opposed to delayed) treatment is likely to be beneficial for avoiding the consequences of chronic *Trypanosoma cruzi* infection. This concept has been previously suggested by other researchers in longitudinal studies with human patients, but it has not been clearly demonstrated in a well-controlled animal model with respect to gastrointestinal disease. The Lewis and Kelly group has pioneered an in vivo imaging system to visualize *T. cruzi* infection in live mice that has allowed them to address questions that were not previously possible.

- Does the work support the conclusions and claims, or is additional evidence needed?

Interestingly, “conclusions” are interspersed in the Results section but are not succinctly stated in the Discussion. It would be helpful if the authors provide a clearer statement of “conclusions” in the Discussion section (like was stated in the Abstract).

One concern with the data relates to the experiments treating mice during the chronic phase of the infection (Fig. 5). There were only 3 mice in the “relapsed” group which weakens their ability to draw conclusions about this group. These data are worth showing, but it might help to make a qualifying comment about that the small size of this one group.

- Are there any flaws in the data analysis, interpretation and conclusions? Do these prohibit publication or require revision?

It would help to consult with a statistician on Figs 1.g. and 5.f. in order show a statistical analysis of these data which does not appear to have been run.

Otherwise, I do not see important flaws in the data analysis or interpretations.

There are two minor points to help the reader: 1) Line 195: the authors refer to “untreated” mice, but the figure refers to this group as “infected”. That can be confusing. 2) Lines 561-2: I believe they have mixed up the meaning of the “lighter colored shaded dots” and the “darker shaded dots”.

- Is the methodology sound? Does the work meet the expected standards in your field?

No concerns about methodology

Reviewer #2 (Remarks to the Author):

The strengths of this study include that the authors present data for the transience in the pathologic impact of *T. cruzi* infection on the mouse gut, with partial reversibility by early (6 weeks post-infection) and less so by later (24 week) anti-parasite drug treatment. Some of the impact of infection and drug treatment-induced recovery is on the apparent replenishment of neuronal tissues/structures. This is particularly interesting given that evidence of neuroregeneration is quite limited.

Data-wise, there is relatively little new or unexpected here, although there are some data that address some of the controversies around Chagas disease. In particular, and as expected and in keeping with most other available data, early treatment resulting in apparent parasitological cure has clinical benefit for gut function. The longer one waits to treat (in this model and also in others) the less reversible is the damage done by this infection. Although not new, this information is important to emphasize in supporting the need for diagnosing and treating *T. cruzi* as early as

possible, but even if that means in the chronic state of infection. This point is frequently ignored as a foundation for preventing Chagas disease. Additionally, these results challenge the existing dogma that neuronal damage is occurring in the gut primarily during the acute infection and this outcome establishes the course of gut dysfunction that ultimately leads to the pathology that accompanies those who develop gut megasyndrome, a minority of those with chronic Chagas disease.

On this latter point, the Discussion seems to muddle the point a bit, noting that (line 264) "Most denervation occurred in the acute phase of infection..." and "Over time, chronic infection of the GI tract led to further neuron losses and gradual decline in GI function." This could be true but I am less sure that the data provided lead directly to this conclusion. The data in Fig 2b suggest that gut dysfunction recovers substantially between 6 and 12 weeks of infection, despite no treatment, and declines again as the infection gets more chronic (the data in Fig 3 did not look at neuronal recovery associated with this functional recovery at week 12). Can the data provided rule out that tissue destruction and replacement (whether it is neuronal or other tissues) are continuous processes but that regeneration just cannot keep up with tissue loss, unless there is drug intervention or near complete immune control of parasite numbers?

Also related to this point, in multiple places, the authors describe DCD as an "enteric neuropathy" and refer to the "central role of denervation in human DCD" but their data do not seem to fully support this view. This enteric neuropathy idea is an hypothesis that seeks to explain gut mega syndromes and is based upon observational data (decrease in neurons in tissues) and has never been addressed except as an association – there are no data that addresses the cause/effect relationship. And the data here also is only associative and in fact seems to argue that at least some of the aspects of this model (i.e. denervation early in the infection) are not causally related to mega syndrome. A clearer treatment of these issues (and I am not doing it justice here) would enhance the impact of the current work. The authors admit that "the disease aetiology is more complex than anticipated because neither parasite load, nor the degree of denervation directly predicted the severity of functional impairment." I question whether they have sufficient data on parasite load and denervation to make this statement and do not recall seeing this analysis done in the study (regressing parasite load or denervation vs functional impairment in individual animals). Although the transcriptional analysis in the mouse gut is a major part of the work, the new information obtained from this is not substantial and is in keeping with the multiple other analyses of tissues infected with *T. cruzi*. Nevertheless, this appears to be the first analysis of this type in gut in mice with chronic *T. cruzi* infection and supports the regeneration model proposed, so it is very useful in that respect.

The authors reiterate their conclusion from previous studies that gut is "a long-term permissive site" for *T. cruzi*." (line 282). The initial support for this in the literature was weak and not supported by work from many other groups. And indeed Fig 1 f and g show that in the 42 week infected mice, there is a substantially higher parasite load in heart and skeletal muscle than in the gut tissues. The authors should address this point.

I have substantial concerns about the "cure" and "relapse" terminology and how this is determined. No doubt they likely have some animals that were cured of the infection and some in which treatment failed. But I question whether the authors can make this discrimination based upon monitoring of luminescence either in whole animals or in dismembered tissues. I recall that prior work from this group indicated that the limit of detection of luminescence was at best ~10 parasites. Documenting treatment failure is clear/easy but documenting cure requires greater sensitivity than that. The use of the term "relapse" also implies that the infection was in some level of remission or in an inactive state prior to the detection of parasites, when in fact it was likely just temporarily suppressed by the treatment – why not just call it treatment failure – that is clearer? The methods used also do not support the discussion of relapse (bottom page 4/top 5) and localization of parasites in particular tissues.

The specific criteria for categorizing mice into cured and relapse groups is not described. Figure 1g seems to show that there are "cured" mice with luminescence levels exceeding those in most "relapse" mice (e.g. there is one "cured" mouse that has a higher parasite detection in the heart than the majority of the "relapse" mice). This figure is incredibly hard to see – the data overlap so as to hide those from the "cured" mice. Starting with Fig 1c and continuing throughout almost the entire paper there is no comparison of difference between the cured and the relapse group in essentially any of the data presented. These statistical comparisons are just not made and based on the data distribution, are unlikely to be statistically different. Fig 1e type graphs should be plotted for all groups, not just the relapse.

Indeed the only data for which there appears to be a visible differential responses in is in Fig 3d which is not surprising as the "relapse" mice should have a profile more like the infected and the majority of the "cured" likely are parasite-free – at least in the gut tissue.

The GI transit time assay is interesting. Text indicates a differential outcome in transit time between the cured and the relapse groups – but statistical support for this difference is not given (in 2c "cured" is compared to infected and "relapse" is compared to uninfected). The end of study constipation phenotypes noted in lines 117-119 are not supported by statistical analysis and the distribution of data suggest that the statement indicating difference in the cured and relapse groups is not true.

Overall, the authors should be more clear about the differences between cured and not-cured – if such differences exist. I don't believe based on the data that they can conclude all the "cured" are indeed "cured". The study would not lose any impact if the groups were renamed "treatment failure" and "apparent success" and not try to tease out difference between them – there appear to be few and those that exist are not worth focusing on for the purposes of this study.

Minor

Fig 2a doesn't really seem necessary as this is a fairly simple experiment

In Fig 1e the authors appropriately mark the "ex-vivo"flux data at the end of study as such , but connect these to the lines showing whole body imaging. Since these ex vivo data are not continuous with the other data points, it would seem more appropriate not to connect these in this way.

Reviewer #3 (Remarks to the Author):

This study by Kahn et al. explores the use of benznidazole, an anti-parasitic chemotherapy, in a mouse model of Digestive Chagas disease (DCD) that the group established previously (PMID: 34424944). Here, the authors provide new insights into an important question in the field, whether DCD is a consequence of acute *T. cruzi* inflammation-induced ENS damage or whether long-term, local, persistence of the *T. cruzi* parasite leads to disease development. In addition, the authors test whether benznidazole treatment during early stages of infection, even in asymptomatic cases, should be treated, thus preventing DCD, compared to chronic phase treatment, as it is routinely done in humans. This study shows new findings that support the concept that the earlier anti-parasitic chemotherapy is started, the greater the chances of preventing or delaying the progression of DCD. However, the underlying mechanisms are not well described.

The authors show that treatment of mice with benznidazole cures the disease, in some cases, while it relapses in others. Control, infected, cured, and relapsed animals are comprehensively analysed at 36 weeks after infection, whereby the authors notice changes in the enteric nervous system (ENS), which the authors describe using terms such as "denervation" and "regenerative neurogenesis". However, those changes in the ENS are not characterised in depth, the actual data presented do not support those statements and the experimental evidence which would justify using those terms are lacking. Therefore, without these mechanistic insights, the study is purely descriptive, and the novelty and significance of the results with regards to ENS-related impact during CDC and after anti-parasitic chemotherapy is compromised. Furthermore, it seems that the study is not fully completed, as the tissue pathology, in particular with regards to the ENS, during chronic phase treatment (Fig. 5) is not characterised, although the authors claim that there is "enteric neuromuscular tissue damage and dysfunction". This is particularly important, since the authors suggest that the infection can be cured at this stage, yet the symptoms remain.

Major comments:

ENS "denervation" and "regenerative neurogenesis":

The authors quantify the number of HuC/D+ expressing neurons in the myenteric plexus of the colon and found a reduced signal after infection. Based on this observation, the authors suggest that there is neuronal loss and ENS degeneration and denervation. HuC/D is not merely a pan-neuronal marker, but its subcellular localization also reflects the condition of a neuron at the time of fixation (PMID 24861242). From our own experience, we know that under certain conditions (such as inflammation, infection, tissue stress) neurons down-regulate or fail to express HuC/D,

however, when assessing DAPI signal, the neurons are still present and not dead/ dying. To provide evidence that ENS degenerates and neurons are dying the authors must assess the impact of infection on the neuronal network (in addition to neuronal cell bodies) and neuronal cell death. Likewise, there is insufficient evidence to support the statement "substantial recovery of myenteric neuron density" [line 123-130] and "regenerative neurogenesis", when there is a possibility that neurons simply re-express HuC/D after resolution of inflammation. For the authors to claim that neurogenesis takes place, they need to show the expression of neurogenesis-related genes, or cells going through proliferation, in addition to lineage tracing to support statements such as: "the data suggest that glial cells are likely to be involved" [line 304-305].

ENS function:

The authors use a total transit time assay to investigate whether the gastrointestinal function and the ENS is affected by infection. While this assay allows for the same mice to be analysed repeatedly over time, it does not specifically allow for ENS-specific function to be analysed, as the GI tract remains connected to brain tissue, receiving commands from CNS. The authors should use an ENS-specific assay to test for ENS-impaired function such as colonic migrating motor complexes or expulsion assay.

Glial cell activity:

It is not clear what the authors mean glial cell activity? In the field, an increase in GFAP+ glial cell reactivity has been described during infection/ inflammation, whereas here, the authors seem to link GFAP+ reactivity to increased glial activity during regeneration. More explanation is needed. Importantly, the dynamics of GFAP need to be accurately quantified as the western blot analysis for GFAP levels, does not seem appropriate (see comments below). Instead GFAP reactivity can be used to quantify glial morphology and GFAP expression levels using image analysis (PMID: 30116771). Also, glial cell proliferation should be analysed.

Mouse model of CDC:

- Can the authors make more comments on the robustness of the mouse model of CDC given that the model has been established by the same group and the original paper has only been cited four times since. Do other groups use the same model? Does the model reflect the clinical manifestation of the disease in human, i.e. only 1/3 of infected mice would be symptomatic [lines 48-50]?
- Does the model mimic the clinical symptoms, and pathology accurately? It is not clear whether neurons are actually lost [lines 61-62], see comment above.
- How does the benznidazole treatment schedule used here in mice compares to that in human [lines 41-43]? Could a difference in dosing regime/ drug concentration explain why the observed sterile parasitological cure rate was higher during the chronic phase of infection as in general there is lower parasite burden [lines 236-238]?
- What are the side effects of benznidazole treatment in mice? Side effects in humans are one of the highest causes for patients stopping the treatment.
- How did the authors assess parasite cure? Was this done purely with the BLI technique? In patients with CDC, parasitic load is usually analysed in blood samples.
- Does the model recapitulate any other symptoms of CDC such as megacolon? Can these symptoms be rescued with benznidazole treatment?
- How is the organisation of the ENS in the mucosa in this model?

Minor comments:

[line 21] The use of the term "sub-curative" treatment in the abstract is misleading, as it gives the impression that some mice were treated with a lower dose of benznidazole.

[lines 22-25] What do the authors mean by "neuro-immune gene expression profiles"? Can those genes be properly highlighted in the results section and figures? Similarly, what do the authors mean by a tissue repair signature? Did the authors perform unbiased gset enrichment analysis which revealed a tissue repair signature after cure?

[lines 95-96] "Of these, 5 (19 %) infections were only detectable by post-mortem ex vivo

96 imaging of internal organs (Fig. 1e).” Is this information really in figure 1e?

[line 159; lines 181-182]: References to figures are missing.

[lines 174-175] The pathway information cannot be found in EDF4?

[lines 179-181] Should this not be Fig 3c instead of Fig. 3e?

[line 194] This should read relapsed instead of relapse

[line 259-261] The authors suggest that asymptomatic should be treated. It would help the reader to highlight how those cases would be identified given that the disease is endemic and the individuals show no symptoms?

Figures:

[Figure 1]

a: Is faecal analysis an established read out for constipation?

d: Include statistics and perhaps reduce the dot size to better appreciate differences between treatment groups.

[Figure 2]

e. and EDF1e: Is the faecal weight plotted from individual faecal pellets or has the weight been averaged and normalised to one pellet from a group of pellets?

[Figure 3]

a: Can some of these genes up/downregulated in each condition be highlighted on the volcano plots?

[Figure 4]

a: Label for the marker (nNOS) in the figure is missing.

d: A GFAP band at 75 kDa has not previously been observed in the gut. Thus, a positive control (such as brain GFAP) should be included to ensure that the antibody is working as expected. In our hands, we never see mouse gut GFAP at 75 kDa. Also, although GFAP degradation has been shown to occur (PMID: 25975510), the resulting band sizes are smaller.

e: Figure is not clear.

[Figure 5]

A similar schematic as in Fig. 1a should be included.

a: How many control mice were used? This information is missing from figure legend.

Extended data figures (EDF)

[EDF1]

Panel h is not clear and needs labelling.

[EDF2] There is no reference to panel 2d.

[EDF3] Panel b – how do the changes in Tuj1 mRNA expression reflects the protein levels in figure 4c?

[EDF4] Panel b, Scale bar too close edge

Methods

Why were only female mice used?

Have the ARRIVE guidelines been followed?

NCOMMS-23-00650-T (Khan *et al*) – Authors’ responses to reviewers’ comments

Our responses are in blue text and the line numbers given correspond to the ‘clean’ version of the revised manuscript. Ahead of our point-by-point responses, we provide a summary of the principal new experiments and analyses that we have done to strengthen our conclusions:

1. We implemented stricter criteria for treatment success and failure and clarified our definitions of cure and relapse in the context of the limit of parasite detection. We then re-analysed every affected data. There were some minor changes in p values for some data sets, but no impact on any of the conclusions.
2. *Ex vivo* parasite load measurements are now presented with clearer thresholds and statistical comparisons.
3. We have developed the strength of evidence for enteric neuronal damage and loss using Hoechst33342 staining of nuclei, TUNEL assays of apoptotic DNA damage and immunofluorescence analysis of the apoptosis marker cleaved caspase 3.
4. We conducted colon contractility analysis to evaluate enteric neuromuscular function in the absence of connection to the CNS.
5. The GFAP Western blot analysis was repeated with a brain tissue control.
6. We carried out new *in vivo* pulse-chase experiments using the thymidine analogue EdU to identify cellular proliferation events following anti-parasitic treatment of *T. cruzi* infection, strengthening our conclusions regarding regeneration of the ENS.
7. We analysed neuropathology at additional stages of infection (3, 12 and 48 weeks), which enabled us to strengthen conclusions about the kinetics of disease pathogenesis and recovery.

Reviewer #1 (Remarks to the Author):

- What are the noteworthy results?

The manuscript by Kahn *et al.* describes novel findings in the murine model of *Trypanosoma cruzi* infection that advance our understanding of the pathogenesis of Chagas disease. Specifically, the data relate to the effects of *T. cruzi* infection on the enteric nervous system and “digestive Chagas disease” (DCD). They show that curative treatment with the drug, benznidazole, leads to the recovery of normal GI function if administered early enough (6 weeks post-infection). Whereas later treatment (24 wks post-infection) result in only partial reversal of DCD. The research includes gene expression profiling and immunofluorescence studies to characterize the effects of the treatments on immune and neural pathways.

- Will the work be of significance to the field and related fields? How does it compare to the established literature? If the work is not original, please provide relevant references.

The work has the important implication that early (as opposed to delayed) treatment is likely to be beneficial for avoiding the consequences of chronic *Trypanosoma cruzi* infection. This concept has been previously suggested by other researchers in longitudinal studies with human patients, but it has not been clearly demonstrated in a well-controlled animal model with respect to gastrointestinal disease. The Lewis and Kelly group has pioneered an *in vivo* imaging system to visualize *T. cruzi* infection in live mice that has allowed them to address questions that were not previously possible.

- Does the work support the conclusions and claims, or is additional evidence needed?

Interestingly, “conclusions” are interspersed in the Results section but are not succinctly stated in the Discussion. It would be helpful if the authors provide a clearer statement of “conclusions” in the Discussion section (like was stated in the Abstract).

1. We appreciate the feedback and have now summarised our key conclusions at start of the discussion.

One concern with the data relates to the experiments treating mice during the chronic phase of the infection (Fig. 5). There were only 3 mice in the “relapsed” group which weakens their ability to draw conclusions about this group. These data are worth showing, but it might help to make a qualifying comment about that the small size of this one group.

2. We did originally include the caveat “Although there were only 3 relapse cases...” when first referring to them, but we accept that it would have been better to be more explicitly cautious. We have now re-organised that section of the results so that all the text relating to the relapse group is unified, starting at line 302 and beginning: “The small

number of relapse cases ($n = 3$) limited our ability to infer the consequences of treatment failure in terms of disease development.”

- Are there any flaws in the data analysis, interpretation and conclusions? Do these prohibit publication or require revision?

It would help to consult with a statistician on Figs 1.g. and 5.f. in order show a statistical analysis of these data which does not appear to have been run.

3. We have taken several steps to improve our analysis and presentation of these *ex vivo* parasite load data:

- The data are now presented as individual organ and tissue types, with GI tract, heart, skeletal muscle and skin retained as the most relevant data in figures 1 and 5 (now figures 1 and 7). The other organs and tissues are now shown in Figures S1 and S9.
- The spread of data for the uninfected control group is now shown to demonstrate the variation in background auto-luminescence signals.
- A threshold line has been added, which is derived from analysis of an internal control, empty ROI, which estimates the level to which bioluminescence signals can ‘spill over’ from one part of the imaging dish to another.
- Statistical comparison of the data (one-way ANOVA) is now shown.

Otherwise, I do not see important flaws in the data analysis or interpretations. There are two minor points to help the reader:

1) Line 195: the authors refer to “untreated” mice, but the figure refers to this group as “infected”. That can be confusing.

4. Agreed, this could be improved. We have:

- edited the text so that this group is now always referred to as “untreated infected” / “untreated infections”;
- added more detail in the starting paragraph of the results to highlight our approach to group terminology (e.g. Line 93: “infected mice administered the vehicle alone, hereon “untreated” infected mice”);
- edited the figure legends to include more detail on the group names.

2) Lines 561-2: I believe they have mixed up the meaning of the “lighter colored shaded dots” and the “darker shaded dots”.

5. This has now been corrected (line 767)

- Is the methodology sound? Does the work meet the expected standards in your field? No concerns about methodology

Reviewer #2 (Remarks to the Author):

The strengths of this study include that the authors present data for the transience in the pathologic impact of *T. cruzi* infection on the mouse gut, with partial reversibility by early (6 weeks post-infection) and less so by later (24 week) anti-parasite drug treatment. Some of the impact of infection and drug treatment-induced recovery is on the apparent replenishment of neuronal tissues/structures. This is particularly interesting given that evidence of neuroregeneration is quite limited.

Data-wise, there is relatively little new or unexpected here, although there are some data that address some of the controversies around Chagas disease. In particular, and as expected and in keeping with most other available data, early treatment resulting in apparent parasitological cure has clinical benefit for gut function. The longer one waits to treat (in this model and also in others) the less reversible is the damage done by this infection. Although not new, this information is important to emphasize in supporting the need for diagnosing and treating *T. cruzi* as early as possible, but even if that means in the chronic state of infection. This point is frequently ignored as a foundation for preventing Chagas disease.

6. We acknowledge that our data add to previous evidence that the earlier anti-parasitic treatment is initiated the higher the likelihood of delaying or preventing the development of disease. However, this body of evidence is fairly small, has so far centred on negativisation of parasitaemia and/or heart disease outcomes, and has not been sufficient to support a clear consensus on whether adults with chronic, asymptomatic infections should be treated (1). We highlight that there have been no studies (human or animal model) that have explicitly tested this hypothesis in the context of digestive disease, and this is where we believe our work offers data that are new, timely and important.

Additionally, these results challenge the existing dogma that neuronal damage is occurring in the gut primarily during the acute infection and this outcome establishes the course of gut dysfunction that ultimately leads to the pathology that accompanies those who develop gut megasyndrome, a minority of those with chronic Chagas disease.

On this latter point, the Discussion seems to muddle the point a bit, noting that (line 264) “Most denervation occurred in the acute phase of infection...” and “Over time, chronic infection of the GI tract led to further neuron losses and gradual decline in GI function.” This could be true but I am less sure that the data provided lead directly to this conclusion. The data in Fig 2b suggest that gut dysfunction recovers substantially between 6 and 12 weeks of infection, despite no treatment, and declines again as the infection gets more chronic (the data in Fig 3 did not look at neuronal recovery associated with this functional recovery at week 12).

Can the data provided rule out that tissue destruction and replacement (whether it is neuronal or other tissues) are continuous processes but that regeneration just cannot keep up with tissue loss, unless there is drug intervention or near complete immune control of parasite numbers?

7. We agree this interpretation was imprecise without data from more time points. The additional experiments have allowed us to add neuron counts for 12 weeks post-infection (Figure S4c) as well as 48 weeks post-infection (Figure 8e). This shows that there is no recovery in neuronal density between 6 and 12 weeks in the untreated infections despite the transient improvement in transit time. This supports the original conclusion about the kinetics of neuron losses being mostly acute, and the phenotype of the BZ-cured mice also shows that ongoing chronic infection “led to further neuron losses and gradual decline in GI function”. We added a clarification to the preceding sentence noting that by “early chronic” we meant the 6 to 12 week period (line 358) so that it would be clear that “Over time, chronic infection...” meant from 12 weeks onwards.

The reviewer’s suggestion that damage and replacement are continuous processes etc, is close to what we believe is probably happening, however, even with the new data we cannot prove this. However, it is interesting to note that nerve growth factor was one of only five genes that were significantly upregulated in all the groups compared to uninfected controls (i.e. untreated, treated cured, treated relapsed), and the only one at $p < 0.01$. Some upregulated genes with ENS functions were shared amongst cures and relapses e.g. *Nefl*, *Plp1*, *Nos1*, whilst others were unique to the cures e.g. *Gfap*, *Rbfox3* (NeuN) (Table S2). Thus, some repair probably is occurring in active infections, but full engagement of an ENS repair programme may indeed be dependent on complete (or near complete) clearance of the infection. We have added some discussion to cover this (lines 375-382).

Also related to this point, in multiple places, the authors describe DCD as an “enteric neuropathy” and refer to the “central role of denervation in human DCD” but their data do not seem to fully support this view. This enteric neuropathy idea is an hypothesis that seeks to explain gut mega syndromes and is based upon observational data (decrease in neurons in tissues) and has never been addressed except as an association – there are no data that addresses the cause/effect relationship. And the data here also is only associative and in facts seems to argue that at

least some of the aspects of this model (i.e. denervation early in the infection) are not causally related to mega syndrome. A clearer treatment of these issues (and I am not doing it justice here) would enhance the impact of the current work.

The authors admit that “the disease aetiology is more complex than anticipated because neither parasite load, nor the degree of denervation directly predicted the severity of functional impairment.” I question whether they have sufficient data on parasite load and denervation to make this statement and do not recall seeing this analysis done in the study (regressing parasite load or denervation vs functional impairment in individual animals).

8. There are several points to make here:

In the introduction (paragraph at line 70) we attempted to summarise the uncertainties around DCD pathogenesis and specifically highlighted the limitations of the existing data to address cause and effect e.g.

“DCD is thought to stem from collateral damage to enteric neurons caused by anti-parasitic inflammatory immune responses” ... “Molecular detection of *T. cruzi* DNA and inflammatory infiltrates in ... human DCD circumstantially suggests that chronic parasite persistence may contribute to disease development ... These data from late and terminal disease states are difficult to interpret in respect of relationships between pathogenesis and infection load or distribution over time.” With this in mind, we have edited the phrase picked up by the reviewer “central role of denervation in human DCD” to now read “relevance of denervation to human DCD” (Line 151)

Chagas heart disease is commonly referred to as a type of cardiomyopathy, due to hallmark myocardial damage, even though many infected people die from sudden cardiac arrests that more likely have a neurogenic than muscular origin. Enteric neuropathy is very clearly a feature of digestive Chagas disease – damage and destruction of the ENS tissues is an indisputable hallmark of the disease. That does not mean this is the only type of pathology present and, as the reviewer highlights, it does not necessarily mean the ENS-specific tissue pathology is necessary and sufficient to cause the symptoms. In terms of cause and effect, our study does address two related mechanistic questions: 1) Does *T. cruzi* infection cause DCD? Answer – Yes; 2) Is acute *T. cruzi* infection sufficient to cause chronic disease? Answer – No.

Whether in humans or animal models, the multi-factorial nature of pathology in *T. cruzi* infections will prevent a formal proof that denervation alone is causative of dysperistalsis and megasyndromes - there is no way to dissociate damage to the ENS from confounding factors such as local and systemic inflammation, smooth muscle damage and cardiac dysfunction. In addition, mice do not develop overt mega comparable to the human disease. Nevertheless, the similarities between Chagas megacolon and congenital megacolon (Hirschsprung disease), which is caused by developmental failure of the ENS to form correctly, are striking (2-4).

As to our statement that “the disease aetiology is more complex than anticipated because neither parasite load, nor the degree of denervation directly predicted the severity of functional impairment.” We have now provided correlation plots (Figure S2e and Figure S3d-e) that back this up and cite data on parasite load vs transit time from our previous paper (Line 366-8).

Although the transcriptional analysis in the mouse gut is a major part of the work, the new information obtained from this is not substantial and is in keeping with the multiple other analyses of tissues infected with *T. cruzi*. Nevertheless, this appears to be the first analysis of this type in gut in mice with chronic *T. cruzi* infection and supports the regeneration model proposed, so it is very useful in that respect.

9. We find it reassuring that the colonic immune response profiles included markers of well-established components of *T. cruzi* infections, e.g. multiple CD8+ T cell related genes, because it provided confidence in the nanostring approach. We would point out that a large majority of prior studies have pre-selected a small number of immune cells/proteins of interest to focus on discrete hypotheses. However, there has only been a handful of low bias screens of host tissue gene expression: microarray analyses of human hearts (5-7), transcriptomes of human blood (8) and placentas (9), and murine hearts (10) and placentas (11). Though there are some broad commonalities in the results, comparison of these studies shows there are also abundant differences in host gene expression that are context-dependent e.g. tissue type, stage of infection.

We are confident that our study is the first to study immune response transcriptional profiles in chronic *T. cruzi* gut infections. We also believe it is the first to integrate such profiles with analysis of nervous system genes in any organ and the first to directly compare them in the context of treatment success and failure. The way we presented the data perhaps did not fully reflect its scope, because we only highlighted a small fraction of the total number of genes that were analysed in the text and figures. The new version now comes with a supplementary table (Table S2) containing the complete gene expression dataset for all 508 genes.

Of note, we have found a substantial number of differentially expressed genes that have not previously been linked to Chagas disease. We highlighted at line 201 “Consistent with the enduring capacity of *T. cruzi* to survive in

this inflammatory environment and the need to prevent excessive GI tissue damage, the up-regulated DEG set included a diverse range of immuno-inhibitory mediators: Btl1, Btl2, Cd274 (PD-L1), Socs1, Lirb4, Lirb3, Lair1, Tnfrsf1, Serpin1." PubMed searches for (cruzi OR chagas) AND (gene name OR protein name) produced 0, 0, 14, 2, 0, 0, 1 and 0 results respectively, so we think this part of the study does identify ways for the field to move in new directions.

The authors reiterate their conclusion from previous studies that gut is "a long-term permissive site" for *T. cruzi*." (line 282). The initial support for this in the literature was weak and not supported by work from many other groups. And indeed Fig 1 f and g show that in the 42 week infected mice, there is a substantially higher parasite load in heart and skeletal muscle than in the gut tissues. The authors should address this point.

10. The chronic GI persistence identified in our original study, in BALB/c mice with CLBR strain (12), has been replicated many times in follow up papers, including expansion to a range of other mouse and parasite strains (13-17). We do not detect much, if any, controversy associated with the data on long-term GI persistence where other authors have cited these papers. Other groups have independently reproduced the observation in papers that we cited in the original ms (18, 19) and we have now added some more (20-23).

Our and others' work has shown that while the GI tract is an important chronic reservoir site, it is not the only one. For example, the skin is infected at a similar rate in multiple models (24). As to the reviewer's point about figure 1f and 1g, it is correct that the heart has a higher infection intensity than the GI tract. The skeletal muscle in fact has a slightly lower average parasite load, though the variance is higher because there were a few animals with very strong signals in this tissue.

We previously showed that the model we are using (female C3H/HeN mice infected with TcI-JR parasites) has a broadly disseminated infection, with high rates of chronic parasitism in many organs (16). This is quite distinctive from the BALB/c and C57BL/6 models that we have worked with, in which infections in sites other than the gut and skin are more rare. In the original manuscript, we clearly acknowledged "the broadly disseminated pattern seen in the infected group" and specifically highlighted that the heart "was a site of frequent and high intensity parasitism in the untreated group"; this information is now at lines 103-108, in slightly revised terms (see point 13 below).

Nevertheless, we do accept that in this manuscript, where we are only using the high dissemination C3H model, it could appear incongruous to focus on the gut as a long-term permissive site. We have therefore added a sentence to the discussion to provide the broader context of data from other models and to emphasise that the data come from mouse studies only. Line 384: "While the TcI-JR-C3H/HeN model is characterised by widely disseminated chronic infections, in several others (e.g. BALB/c, C57BL/6) *T. cruzi* is mainly restricted to the stomach, colon and skin. Why the GI tract serves as a long-term permissive site for *T. cruzi*, at least in mice, has been unclear...".

I have substantial concerns about the "cure" and "relapse" terminology and how this is determined. No doubt they likely have some animals that were cured of the infection and some in which treatment failed. But I question whether the authors can make this discrimination based upon monitoring of luminescence either in whole animals or in dismembered tissues. I recall that prior work from this group indicated that the limit of detection of luminescence was at best ~10 parasites. Documenting treatment failure is clear/easy but documenting cure requires greater sensitivity than that.

11. We previously estimated it to be <20 parasites for the bioluminescent CLBR strain, which expresses luciferase at a similar level to the JR strain used here (16, 24). Therefore, it is true that we cannot rule out the presence of extremely low numbers of parasites persisting in animals that we termed "cured", a problem that is common across the infectious diseases field where it is not possible to prove there are zero individual microorganisms in a whole animal or patient.

The only established method to realistically achieve "greater sensitivity than that" for *T. cruzi* in mice would have been to administer immunosuppressive drugs to promote outgrowth of the hypothetical residual parasite populations that are so tightly controlled by the immune system. However, this approach would have confounded all of our other analyses of host response, pathology and organ function.

Nevertheless, we do accept that greater rigour and clarity is required regarding our use of the terms cure and relapse, and we have taken steps to address this:

We originally based cure and relapse calls on the observation of "at least 10 contiguous bioluminescent pixels of radiance $\geq 3 \times 10^3$ photons/second/cm²/sr" based on previous work (25). However, prompted by the reviewer's justified scrutiny of the *ex vivo* imaging data, we investigated whether a more appropriate threshold could be

established. Looking again at the raw data, we found that the level of background noise varied between our two *in vivo* imaging systems (Lumina II and Spectrum) and shifted slightly after routine servicing events. We also looked more systematically at uninfected control images to investigate the variability in auto-luminescence associated with different tissue/organ types, for example, a) identifying weak signals co-localising with small intestinal lumen contents giving them a higher threshold than the global average, b) finding substantially lower than average auto-luminescence for heart samples, giving them a lower threshold than the global average, c) noting that auto-luminescence (false positive) signals differed in their scatter patterns compared to those derived from *T. cruzi* infection foci, which characteristically radiate in intensity out from a discrete point source of light emission. Figures 1 and 2, below, summarise these issues.

Figure 1. Background auto-luminescence levels vary between different organs taken from uninfected control mice. Data shown as total flux and radiance (total flux corrected for size differences).

Figure 2. Differentiating infection-focus bioluminescence from intestinal lumen-associated auto-luminescence. Top row shows images for a borderline relapse/cure call that was finally assessed as relapsed. Bottom row shows images for an uninfected control. Each series shows the same data with an increasingly strict cut-off (left → right) for the pseudocolour heat map that is overlaid on the photograph. Signals that radiate from a point source (green arrows) are interpreted as compatible with a *T. cruzi* infection focus. Signals that are co-localising with gut contents (white dashed ellipses) are interpreted as background. Note the mice were fasted before necropsy and there is greater retention of food/faeces in the infected example than the uninfected control.

We therefore re-analysed all of the *ex vivo* imaging data as follows:

Background-matched controls were used to normalise the fold-change bioluminescence radiance calculations e.g. infected sample images taken on the Spectrum instrument before and after servicing were analysed using reference (uninfected control) data from equivalent runs rather than a global average.

We then implemented a two-step process to call cures (apparent treatment success) and relapses (treatment failure) more rigorously. Firstly, any BZ-treated animal for which at least one sample had a fold-change radiance measurement greater than 2.49 (equal to the highest value seen in the uninfected control group) was scored as relapsed. Secondly, the *ex vivo* images for all remaining BZ-treated animals were manually inspected for the presence of discrete bioluminescence foci for which the signal radiated from a point source as the threshold was gradually reduced to noise (see Figure 2 above). Compared to the original submission, we detected two cases of assessed treatment failure (relapses) that had originally been called as cured. We now provide a supplementary data file (Table S1) detailing all of the radiance values for every tissue/organ for every animal, annotated with the final cure vs relapse calls.

All downstream analyses were then repeated with the revised groupings. Importantly, the overall outcome of this re-analysis was limited to minor shifts in *p* values, with no change in our conclusions.

Our revised definitions for cure and relapse are now explicitly set out in a new part of the methods section entitled "Treatment Outcomes" (line 503). We also included acknowledgement of the limits of detection associated with our approach and attempted to relate our terminology to what has been used in recent Chagas disease clinical trials (26, 27) e.g. in the results section, line 113 "parasitological cure" → "sustained parasite clearance, here considered as parasitological cure"

We thank the reviewer for prompting us to take a more rigorous approach.

The use of the term "relapse" also implies that the infection was in some level of remission or in an inactive state prior to the detection of parasites, when in fact it was likely just temporarily suppressed by the treatment – why not just call it treatment failure – that is clearer?

12. Our data show that there was not a uniform time or spatial pattern in the way that infections rebounded – they were very heterogeneous and often not detected until several months after benznidazole was withdrawn, including many that were only detected by *ex vivo* organ imaging. In our view, the term "treatment failure" does not capture the observed suppression and (heterogeneous) rebound dynamics because it could just as well apply to a situation where the drug had no impact on parasite loads or disease symptoms. There are several precedents for using the term "relapse" to describe the situation when a recurrent infection is detected after suppressive, but non-curative treatment has ended. It provides a useful distinction, e.g. in tuberculosis (28), from re-infection (with a new strain) and rebound of the pre-treatment pathogen population. The term has been used in the same sense that we are using it in recent clinical trials for both visceral leishmaniasis (29) and Chagas disease (26, 27).

As set out in the previous point, we have now provided clear and detailed explanations of our use of the terms cure and relapse in the methods section.

The methods used also do not support the discussion of relapse (bottom page 4/top 5) and localization of parasites in particular tissues.

13. We assume that the relevant section (lines 98-101 in the original ms) read: "*Relapse infections were most often localised to GI tissues, in contrast to the broadly disseminated pattern seen in the infected group (Fig. 1f, 1g). Of note, in the context of cardiac Chagas disease, relapse infections rarely localised to the heart (2/11, 18%), which was a site of frequent and high intensity parasitism in the untreated group (16/18, 89 %) (Fig. 1f, 1g).*"

This section has been revised following the re-analysis of the *ex vivo* data and the implementation of the new cure and relapse definitions outlined in point 11.

The data still show that relapse infections are more often localised to GI tissues than not (Figure 1g and the new Supplementary Table 1). However, we do accept that our original wording did not adequately convey the variability between individual relapses - 8/13 relapses had GI localisation while 5/13 did not. This is not a significant difference, so we removed the statement that "*Relapse infections were most often localised to GI tissues*".

To support our statement that the distribution of relapse infections contrasted with broad dissemination in the untreated infections, we provide a new part to figure 1f showing the number of sites in which parasites were detected per mouse. This demonstrates that relapses are statistically significantly less disseminated ($p < 0.0001$).

To support our point about heart-specific infections, we now report (line 109) that Fisher's exact test shows relapse infections are indeed less frequently localised to the heart: 17/18 untreated vs 6/13 BZ-relapse ($p = 0.0041$) and the new Figure 1g shows that the heart-infection intensities are significantly lower.

Lastly, we also should have highlighted the caveat that these snapshot in time end-point measurements might not reflect the true spatio-temporal dynamics of the infections, which we now do (line 110).

The specific criteria for categorizing mice into cured and relapse groups is not described.

14. This has now been addressed as detailed in point 11. A new subsection entitled "Treatment Outcomes" was added to the methods (line 503).

Figure 1g seems to show that there are "cured" mice with luminescence levels exceeding those in most "relapse" mice (e.g. there is one "cured" mouse that has a higher parasite detection in the heart than the majority of the "relapse" mice). This figure is incredibly hard to see – the data overlap so as to hide those from the "cured" mice.

15. This has now been addressed as detailed in points 3 and 11. The figure panel has been replaced with clearer versions.

Starting with Fig 1c and continuing throughout almost the entire paper there is no comparison of difference between the cured and the relapse group in essentially any of the data presented. These statistical comparisons are just not made and based on the data distribution, are unlikely to be statistically different.

16. These comparisons were made, but for clarity of presentation we followed the convention of only annotating p values < 0.05 for tests with many pairwise comparisons. Thus, the reviewer is correct that there were few significant differences for the direct comparison of the cure and relapse groups. We now state in the legends where only significant differences are annotated on the figures.

There are some further considerations:

- a. Every relapse case is a unique individual event e.g. some infections rebounded much earlier and more strongly than others, some were only barely detectable and only in a non-GI tissue at necropsy. This limits the value of a direct comparisons that group all the relapses together.
- b. It was not a primary aim of our study to directly compare the consequences of sterile cure vs relapse in our study. The experiments were designed and powered on the basis of a predicted 70% reversal of transit time delay after curative BZ treatment. Full details are now given in the statistics section of the methods, now re-titled "Sample size and statistics" at line 669.
- c. Nevertheless, we do think it is both useful and important to consider the differential outcomes of comparisons between of the cure and relapse groups to the healthy, uninfected controls and to the untreated infection controls. On a range of measures the cured mice were not significantly different from healthy controls when the relapses were significantly different from healthy controls. These included:
 - Ex vivo GI parasite loads (Fig 1g)
 - Total GI transit time at endpoint (Fig 2b)
 - Faecal pellet retention after fasting - n pellets and dry weight (Figs 2c,2d),
 - Myenteric plexus neuron density in proximal and distal colon (Fig 3d)
 - A set of 149 differentially expressed genes – 29% of the 508 profiled using nanostring (Fig 4b and 4e vs Figure S 6c)
 - Distal colon myenteric plexus nitrergic neuron density (Fig 5b)
 - Faecal pellet retention after fasting (24 wpi treatment experiment), n pellets and dry weight (Fig 8c)

Of the above measures, those for which cures *additionally* showed a significant normalisation compared to the untreated infected group, but the relapses *did not*, were as follows:

- Ex vivo GI parasite loads (Fig 1g)

- Total GI transit time at endpoint (Fig 2b)
- Myenteric plexus neuron density in distal colon (Figs 3d)
- Another set of 102 differentially expressed genes – 20% of the 508 profiled using nanostring (Supplementary Table 2)

- d. We wish to emphasise that the cured (sustained clearance) mice were the principal focus of our analysis, particularly in the context of ENS recovery. We described our observations relating to relapse events in the results section, but then only mentioned them once in the abstract and once in the discussion, both as counter-factuals to the significant normalisation of transit time in cured mice. This remains an important point to make as it goes some way to meeting the need for an “add-back” type control to address the question of causality. This was important given our specific aim of determining cause and effect with regard to acute vs chronic infection and digestive disease.

Fig 1e type graphs should be plotted for all groups, not just the relapse.

17. We agree this would be useful. These charts showing the bioluminescence/parasite load levels over time for individual mice in the other groups are now provided in supplementary figure 1a.

Indeed the only data for which there appears to be a visible differential responses in is in Fig 3d which is not surprising as the “relapse” mice should have a profile more like the infected and the majority of the “cured” likely are parasite-free – at least in the gut tissue.

18. Figure 4d (formerly 3d) shows only the average for each group, the heat map does not enable visualisation of the spread of data or statistical comparison. When each pathway is presented in a way that shows the spread and statistics, as was done in supplementary figure 3 (now Figure S 5a), the comparison between groups for some (not all) pathways resembles much more closely the patterns seen for other data types i.e. the untreated infected controls are significantly different from uninfected controls while the BZ-cured mice are not. The relapses are also significantly different from the uninfected controls for some pathways. We also feel it is relevant here to highlight the Venn diagrams (Figure 4b), which show individual gene-level analyses, identifying many individual genes for which expression levels differ significantly between cures and relapses. In the new supplementary table 2, we provide *p* values for all genes in BZ-cures and BZ-relapses vs the untreated infected and the uninfected controls.

The GI transit time assay is interesting. Text indicates a differential outcome in transit time between the cured and the relapse groups – but statistical support for this difference is not given (in 2c “cured” is compared to infected and “relapse” is compared to uninfected). The end of study constipation phenotypes noted in lines 117-119 are not supported by statistical analysis and the distribution of data suggest that the statement indicating difference in the cured and relapse groups is not true.

19. The differences between cures and relapses were not significantly different for those data and for clarity of the figure presentation we followed the convention of only annotating *p* values <0.05 for tests with many pairwise comparisons. The *p* values for this comparison were as follows: 0.11 for transit time, 0.27 for faecal pellet counts and 0.46 for faecal dry weight.

However, continuing the reasoning set out above in point 16, we maintain that it is valid and important to draw conclusions based on the finding that the cures are not significantly different to the healthy uninfected controls when the infected untreated (vehicle) controls and the relapses *are* significantly different from those healthy uninfected controls. These differences in significance *vs healthy mice* for the cures and relapses are likely to be biologically meaningful.

We have re-phrased the relevant lines to improve our accuracy:

“This was alleviated in benznidazole cured mice, but not in relapsed infections” → “This was alleviated in benznidazole cured mice to the point that they were not significantly different from uninfected control mice” (Line 132).

Overall, the authors should be more clear about the differences between cured and not-cured – if such differences exist. I don’t believe based on the data that they can conclude all the “cured” are indeed “cured”. The study would not lose any impact if the groups were renamed “treatment failure” and “apparent success” and not try to tease out difference between them – there appear to be few and those that exist are not worth focusing on for the purposes of this study.

20. As detailed above in point 11, we have now clarified our definitions of cure and relapse with acknowledgment of the associated limits of detection.

As detailed in point 16, the frequency of relapses was a secondary outcome of the study, but we do think the distinction compared to cures is useful because of the observed treatment failure rates in human patients. In our view, we did not try to tease out differences unduly between cures and relapses. We described our observations in the results section and only mentioned the relapses once in the abstract and once in the discussion, both as counter-factuals to the normalisation of transit time in cured mice.

Minor

Fig 2a doesn't really seem necessary as this is a fairly simple experiment

21. Agreed, we have removed the schematic from the figure.

In Fig 1e the authors appropriately mark the "ex-vivo" flux data at the end of study as such, but connect these to the lines showing whole body imaging. Since these ex vivo data are not continuous with the other data points, it would seem more appropriate not to connect these in this way.

22. Agreed, we now present these data so that the *in vivo* and *ex vivo* components are on independent y axes and joined by dashed lines to reflect the fact they are not continuous data.

Reviewer #3 (Remarks to the Author):

This study by Kahn et al. explores the use of benznidazole, an anti-parasitic chemotherapy, in a mouse model of Digestive Chagas disease (DCD) that the group established previously (PMID: 34424944). Here, the authors provide new insights into an important question in the field, whether DCD is a consequence of acute *T. cruzi* inflammation-induced ENS damage or whether long-term, local, persistence of the *T. cruzi* parasite leads to disease development. In addition, the authors test whether benznidazole treatment during early stages of infection, even in asymptomatic cases, should be treated, thus preventing DCD, compared to chronic phase treatment, as it is routinely done in humans. This study shows new findings that support the concept that the earlier anti-parasitic chemotherapy is started, the greater the chances of preventing or delaying the progression of DCD. However, the underlying mechanisms are not well described.

The authors show that treatment of mice with benznidazole cures the disease, in some cases, while it relapses in others. Control, infected, cured, and relapsed animals are comprehensively analysed at 36 weeks after infection, whereby the authors notice changes in the enteric nervous system (ENS), which the authors describe using terms such as "denervation" and "regenerative neurogenesis". However, those changes in the ENS are not characterised in depth, the actual data presented do not support those statements and the experimental evidence which would justify using those terms are lacking. Therefore, without these mechanistic insights, the study is purely descriptive, and the novelty and significance of the results with regards to ENS-related impact during CDC and after anti-parasitic chemotherapy is compromised. Furthermore, it seems that the study is not fully completed, as the tissue pathology, in particular with regards to the ENS, during chronic phase treatment (Fig. 5) is not characterised, although the authors claim that there is "enteric neuromuscular tissue damage and dysfunction". This is particularly important, since the authors suggest that the infection can be cured at this stage, yet the symptoms remain.

Major comments:

ENS "denervation" and "regenerative neurogenesis":

The authors quantify the number of HuC/D+ expressing neurons in the myenteric plexus of the colon and found a reduced signal after infection. Based on this observation, the authors suggest that there is neuronal loss and ENS degeneration and denervation. HuC/D is not merely a pan-neuronal marker, but its subcellular localization also reflects the condition of a neuron at the time of fixation (PMID 24861242). From our own experience, we know that under certain conditions (such as inflammation, infection, tissue stress) neurons down-regulate or fail to express HuC/D, however, when assessing DAPI signal, the neurons are still present and not dead/ dying. To provide evidence that ENS degenerates and neurons are dying the authors must assess the impact of infection on the neuronal network (in addition to neuronal cell bodies) and neuronal cell death.

23. We have conducted new experiments and now use several independent methods to further support the conclusion that *T. cruzi* infection is leading to neuronal damage and death. The results are outlined in the paragraph starting at line 151. To summarise:

- a. We use cleaved caspase-3 immunofluorescence to show that apoptosis executioner caspase 3 is activated inside myenteric ganglia (Figure S4a) of acutely infected mice.

- b. We use TUNEL staining to show widespread apoptotic DNA damage, including TUNEL-positive nuclei within myenteric ganglia of acutely infected mice (Figure 3a)
- c. New images showing Hu expression and Hoechst33342-stained DNA are provided (Figure 3b, 3c, Figure S4d) with annotation highlighting atypical neuron soma morphologies and associated DNA structures, including the presence of pyknotic nuclei as evidence that progression to late stages of the apoptosis pathway is occurring.
- d. We have employed the “ANNA-1” anti-Hu reagent as an alternative to the anti-HuC/D antibody for the new neuron count experiments for 12 and 48 wpi. At these time-points we again found significantly reduced myenteric neuron density in infected mice (Figure 8, Figure S4).

Collectively, these lines of evidence together strongly support our conclusion that *T. cruzi* infection is leading to death of myenteric neurons, as observed in human cases of digestive Chagas disease.

Likewise, there is insufficient evidence to support the statement “substantial recovery of myenteric neuron density” [line 123-130] and “regenerative neurogenesis”, when there is a possibility that neurons simply re-express HuC/D after resolution of inflammation. For the authors to claim that neurogenesis takes place, they need to show the expression of neurogenesis-related genes, or cells going through proliferation, in addition to lineage tracing to support statements such as: “the data suggest that glial cells are likely to be involved” [line 304-305].

24. The strength of evidence detailed above for loss of neurons necessarily also strengthens the conclusion that the generation of new neurons explains the significant recovery in Hu+ neuron numbers after BZ-mediated cure.

From the literature it appears that few if any neurogenesis-related genes have been well validated in the ENS. We have shown upregulation of *Ngf*, *Notch2* and *Zeb1* in bulk colon RNA, which could be considered as “neurogenesis-related genes”, as we discuss (Paragraph at lines 217 and discussion lines 421 – 427). The gene Nestin has been used by Kulkarni et al (30), but they also found Nestin expression in pericytes and Belkind-Gerson *et al* (31) further point out: “In the gastrointestinal (GI) tract, Nestin is expressed by glial cells, interstitial cells of Cajal (ICC), endothelial cells, pericytes, and stromal tumor cells [18]. It has also been found to be expressed within and surrounding myenteric and submucosal ganglia - in cells smaller and more numerous than neurons [19].” We therefore consider that a deeper investigation of the molecular basis of re-innervation in our model is more appropriate for future studies.

Regarding lineage tracing, we agree that this would potentially be very informative. However, the relevant transgenic mice have a C57BL/6 background and our model uses C3H/HeN mice. Therefore, substantial further development i.e. of either a B6 model of DCD or C3H transgenics would be required.

We have been able to investigate cellular proliferation, by using the thymidine analogue EdU. This has revealed that cells proliferating in the weeks after benznidazole treatment give rise to progeny that are present within and around the myenteric plexus ganglia after both a short (1 week) and long (17 weeks) follow-up period. A subset of the EdU retaining cells co-localised with GFAP protein, and in a few rare cases Hu. These data are presented in figure 6 and S8, and in a new results section “Analysis of myenteric cellular proliferation after treatment of *T. cruzi* infection” (Line 264). Our conclusion is that cellular proliferation is clearly a feature of the post-cure tissue repair environment, but the data did not convincingly link this to neurogenesis, and we have revised our conclusions accordingly.

Regarding our statement “the data suggest that glial cells are likely to be involved” – this is a fair point, our phrasing was too imprecise. We have removed this phrase and brought in discussion of the results from the new EdU experiments and highlighted context from a new paper published during the revision period by Laddach et al (32) (see lines 410 – 421).

We also changed “regenerative neurogenesis” in the abstract to “recovery of neuronal density” (line 28).

ENS function:

The authors use a total transit time assay to investigate whether the gastrointestinal function and the ENS is affected by infection. While this assay allows for the same mice to be analysed repeatedly over time, it does not specifically allow for ENS-specific function to be analysed, as the GI tract remains connected to brain tissue, receiving commands from CNS. The authors should use an ENS-specific assay to test for ENS-impaired function such as colonic migrating motor complexes or expulsion assay.

25. We have now included some colon basal contractility analysis obtained from organ bath experiments. We found a significant reduction in contraction frequency in samples from *T. cruzi* infected (untreated) mice that was reversed after BZ treatment. We also measured amplitude, which showed the same trends in terms of group means, but there were no significant differences. Our sample size is small ($n = 3$ for BZ-cures), so we are cautious about making specific inferences about these phenotypes, but the reduction in contractile frequency is consistent with the idea that our other assays of GI function do reflect ENS-intrinsic phenomena. Example traces and quantification of these data are now presented in Figure 2.

Glial cell activity:

It is not clear what the authors mean glial cell activity? In the field, an increase in GFAP+ glial cell reactivity has been described during infection/ inflammation, whereas here, the authors seem to link GFAP+ reactivity to increased glial activity during regeneration. More explanation is needed.

26. Our data do not show evidence of increased GFAP reactivity in untreated infections – the Western blot, IFA, Nanostring and qPCR data all showed this. The new EdU-based proliferation experiments showed slightly higher frequencies of EdU+ GFAP+ co-localisation in infected vs uninfected mice in some cases, but these were not significant differences. Another glial cell marker, S100 β , was significantly downregulated in chronic infections and then upregulated after cure (Figure 4e). This suggests that *T. cruzi* infection affects enteric glial cells in different ways than the infections the reviewers is referring to.

Nevertheless, it is a fair point that our phrase “glial cell activity” was vague. In the abstract (line 27) we changed the phrase “glial cell activity” to “glial cell marker expression”. At line 261, we changed the phrase “EGC activity” to “increased GFAP expression in EGCs and/or proliferation of GFAP+ EGCs”.

As per point 24, the potential link between glial cells and regeneration of myenteric neurons remains circumstantial and is still a hypothesis based on models of adult neurogenesis from other experimental systems and the changes we have seen in GFAP expression after benznidazole treatment. The revised discussion is now clearer on this aspect of the study.

Importantly, the dynamics of GFAP need to be accurately quantified as the western blot analysis for GFAP levels, does not seem appropriate (see comments below). Instead GFAP reactivity can be used to quantify glial morphology and GFAP expression levels using image analysis (PMID: 30116771). Also, glial cell proliferation should be analysed.

27. We have now included a brain tissue control for the GFAP Western blot (Figure S 7), which does run at the typical ~50kD level. Importantly, the immunofluorescence assays use the same primary anti-GFAP antibody (e.g. Figure 5c) and the images show labelling of glial cell networks with excellent specificity. On re-inspection of the original blot images and comparison with new runs, we realised that the 75 kDa ladder band had not shown up under the imaging settings we had originally used and had therefore mistaken the 100 kDa ladder band for the 75. Thus, the slow running colon GFAP bands we observe are in fact close to 100 kDa, with the secondary bands running at around 75 kDa. Therefore, the simplest explanation for the difference between colon and brain (in our hands at least) is dimerization of GFAP (which is covered in the reference provided by the reviewer – see point 49). We have tried stronger denaturation conditions, but this has resulted in disappearance of the colon GFAP bands. The text and figures have been updated with the correct band sizes.

We do not think that these issues affect the utility of the assay for relative quantification of GFAP protein abundance because the intensity of staining is proportional to the amount of antigen present in the sample, independent of the speed at which the protein of interest runs.

We did highlight observations of GFAP patches, aggregates and distinct glial cell morphologies post-treatment (line 247-251) and annotated examples in figure 5c (see “Ganglion 2”).

As detailed above (point 24) we have conducted new cellular proliferation experiments using EdU and found some evidence of increased GFAP+ glial cell proliferation after BZ treatment (Figure 6, Figure S8).

Mouse model of CDC:

- Can the authors make more comments on the robustness of the mouse model of CDC given that the model has been established by the same group and the original paper has only been cited four times since. Do other groups use the same model?

28. It has been known for decades that C3H mice are particularly susceptible to *T. cruzi* infection. We have worked with the parasite and mouse strain combination (Tcl-JR in C3H/HeN) for more than 10 years, but originally

focused on the cardiac form of Chagas disease. The papers from these studies have been cited hundreds of times. The development of the model for digestive disease has been more recent and our original paper now has twelve citations. We are aware that the model is being established in two other UK institutions. We are hopeful that more people in the field will take on the model, however, the field has developed over the years without a strong culture of standardisation of animal models between laboratories – this has its advantages e.g. more biology is explored, but one downside is reduced independent replication. It is also worth noting that exchanging parasite strains is highly restrictive, especially across international borders, due to *T. cruzi*'s BSL3 categorisation.

Does the model reflect the clinical manifestation of the disease in human, i.e. only 1/3 of infected mice would be symptomatic [lines 48-50]?

29. The spread of the transit time data in Figure 2b shows that there is heterogeneity in the severity of dysperistalsis, but there is a highly significant difference compared to healthy controls. The reason for variable clinical outcomes in humans is normally attributed to a combination of genetic (host, parasite) and environmental factors, though there are very limited data on the mechanisms beyond these broad categories. C3H/HeN mice are genetically inbred and we use a clonal parasite strain, so it is not unexpected that reproducibility of the disease phenotype is very high compared to disease rates in infected people. Of note, in our original paper we described screening twelve mouse and parasite strain combinations in order to find one that had a consistently strong and significant transit time delay (13).

Does the model mimic the clinical symptoms, and pathology accurately? It is not clear whether neurons are actually lost [lines 61-62], see comment above.

30. It is important to emphasize that DCD in humans is itself highly variable in its presentation, it is under-diagnosed and clinical data are fragmentary (most data are from late or end-stage disease), but it is clear that the spectrum covers different regions of the GI tract (oesophagus and/or colon, also rarely small bowel or stomach) and different rates of progression from mild symptoms through to full megasyndromes, as well as non-progression. We referred to this in the introduction, but have edited slightly to make it clearer that progression is not inevitable: “Eventually, in some cases, massive organ dilatation results in megasyndromes...” (Line 64).

Our mouse model does not progress to megacolon and does not, macroscopically at least, have enlargement of the oesophagus. This may be because of our relatively short experimental time frame of 5-6 months vs decades in humans, or it may reflect intrinsic physiological differences between the two species. However, megasyndromes are end-stage disease outcomes and therefore less critical for a model to reproduce than the earlier pathogenesis mechanisms, which may be interruptable or reversible. We do see inflammatory infiltrates in the colon smooth muscle wall, as seen in clinical specimens, and as detailed above (point 23) there is strong evidence that enteric neurons are lost. We are therefore probably closer to recapitulating the earlier stages of disease development, or a window on a continuous and cumulative tissue damage process that plays out over a much longer period of time in humans.

We continue to investigate the similarities and differences between human disease and the mouse model.

• How does the benznidazole treatment schedule used here in mice compares to that in human [lines 41-43]? Could a difference in dosing regime/ drug concentration explain why the observed sterile parasitological cure rate was higher during the chronic phase of infection as in general there is lower parasite burden [lines 236-238]?

31. The standard schedule for treating *T. cruzi* infected people is 60 days at 5-10 mg/kg. A recent clinical trial in Bolivia showed that shortening the schedule to 2 or 4 weeks achieved comparable outcomes in terms of parasite clearance (33). Our dosing is adjusted using standard pharmacological multiplication factors for mice (approx. 12x). Our cure/relapse rates are in the same range as observed in clinical trials e.g. the largest trial (BENEFIT) reported parasite clearance in 44% (Colombia/El Salvador), 86% (Brazil) and 73% (Argentina) of participants treated with benznidazole.

Relapses were seen in 13 of 27 mice treated at 6 weeks (48%) vs 3 of 10 mice treated at 24 weeks (30%). Using Fisher's exact test we have found these rates to be not significantly different. We added a line explaining this to the relevant part of the results text (Line 296) “The proportions of cures and relapses was not significantly different from those observed for treatment at 6 wpi (Fisher's exact test $p = 0.461$).”

The lines quoted by the reviewer about parasite load do not seem to correspond to the point made, we presume they meant lines 226-228? In any case, the pre-treatment parasite loads at 6 and 24 wpi differ by almost two orders of magnitude (figures 1c and 7b), so compared with a 1.6-fold (non-significant) difference in relapse rates, it

seems unlikely that parasite load is a major factor. Why not is an interesting question, but following up would be outside the scope of the current study.

- What are the side effects of benznidazole treatment in mice? Side effects in humans are one of the highest causes for patients stopping the treatment.

32. In our experience mice tolerate benznidazole well. In terms of the side effects seen in humans, there are no overt changes in behaviour that might suggest peripheral neuropathy or skin irritation. We also did not observe reduced body weights linked to benznidazole treatment (see Figures S2a and S7b), so we infer there is no substantial change in appetite.

- How did the authors assess parasite cure? Was this done purely with the BLI technique? In patients with CDC, parasitic load is usually analysed in blood samples.

33. Yes, these calls were made based only on BLI data. We previously showed that there is an excellent correlation between ex vivo bioluminescence-inferred and qPCR-inferred parasite loads in tissues samples that were evaluated using both techniques (12). Despite its superior sensitivity, the major limitation of qPCR for calling cures and relapses is the risk of false negatives e.g a blood sample containing no parasites testing negative when there are in fact foci of tissue infection remaining. Accordingly, we previously compared imaging and PCR in the context of drug treatment experiments (benznidazole and posaconazole) that included use of the cyclophosphamide immunosuppression method (see point 11) and found that PCR had a tendency to overestimate the cure rate (17). As detailed in point 11 for reviewer 2, we have clarified our definitions of treatment success and failure.

- Does the model recapitulate any other symptoms of CDC such as megacolon? Can these symptoms be rescued with benznidazole treatment?

34. There was no macroscopic evidence of megacolon - please see point 30 above for further discussion.

- How is the organisation of the ENS in the mucosa in this model?

35. This is an interesting question. We focused much of our study on the myenteric plexus because previous studies indicated that the majority of parasites were located in the muscle wall (24). Some of our data are from full thickness samples e.g. nanostring mRNA profiling and Western blots, which include the mucosal regions, though admittedly not isolated mucosal ENS.

As part of our efforts to address the reviewer's critique about evidence for neuronal loss (point 23) we started to work with colon full thickness transverse sections and, interestingly, we did see evidence of apoptosis (TUNEL staining) in all regions at 3 weeks p.i. (Figure 3a). However, we suggest that a detailed analysis of the mucosal ENS could be most appropriately investigated as part of future research.

Minor comments:

[line 21] The use of the term "sub-curative" treatment in the abstract is misleading, as it gives the impression that some mice were treated with a lower dose of benznidazole.

36. Agreed, this has been changed to "treatment failure"

[lines 22-25] What do the authors mean by "neuro-immune gene expression profiles"? Can those genes be properly highlighted in the results section and figures? Similarly, what do the authors mean by a tissue repair signature? Did the authors perform unbiased gset enrichment analysis which revealed a tissue repair signature after cure?

37. The term "neuro-immune gene expression profiles" is used simply as a shorthand for "nervous system and immune system gene expression profiles" because our nanostring panel is the off-the-shelf mouse immunology panel plus a custom set of probes taken from their neuroinflammation and neuropathology panels. In hindsight we can see that use of a dash as punctuation could potentially be interpreted as related to the concept of neuro-immunity or direct interactions between the two systems. We have therefore changed to "neuro/immune" (line 26) and "nervous and immune system" (line 373) to keep things more separate.

Tissue repair signature was used as a shorthand for a subset of genes known to have roles in tissue repair processes. We did not claim there was statistically significant enrichment as the gene set analysis tool in the nanostring nsolver software package relied on pre-determined pathway annotations for the probes and the set of pathway terms available did not include any that encompassed tissue repair. Use of the shorthand term "signature" was only used in the abstract and the relevant figure legend where brevity was essential; we have now changed "signature" to "profile" in those places; in the main text we stated: "multiple genes involved in tissue repair and regeneration were also highly significant DEGs in cured mice, including *Notch2*, *Tgfb2*, *Tgfb1*, *Tgfb3* and

Zeb1.” (Line 226). The newly included Table S2 will enable interested readers to view the complete dataset, including the in-built nanostring pathway annotation terms. In future work we plan to investigate this area further using genome-scale tools, to reduce the gene selection bias even further and enable the type of enrichment analysis mentioned by the reviewer.

[lines 95-96] “Of these, 5 (19 %) infections were only detectable by post-mortem *ex vivo* imaging of internal organs (Fig. 1e).” Is this information really in figure 1e?

38. Technically the data did show this – the BLI signal did not get above the threshold line until the *ex vivo* data point for those five individuals. However, we accept the data presentation could have been clearer. As per point 22 for reviewer 2, we have now added a second y axis specifically for the *ex vivo* data and also changed the formatting of the data points/lines to highlight these post-mortem relapse calls.

Also note, as per point 11 for reviewer 2, two mice were re-assigned from cure to relapse, so it is now 7 (26%).

[line 159; lines 181-182]: References to figures are missing.

39. Call outs for Figure 4a, 4e and Table S2 have now been added to the text.

[lines 174-175] The pathway information cannot be found in Figure S4?

40. Apologies, the call out was incorrect, it should have been Figure S3, which is now Figure S5.

[lines 179-181] Should this not be Fig 3c instead of Fig. 3e?

41. Both panels are relevant here and are now called out in the text. We thought the coordinates of plot 3e (now 4e) showed the contrast in S100b expression more precisely than the colour shades in 3c – significant downregulation in infected, significant upregulation in cures.

[line 194] This should read relapsed instead of relapse

42. This has been corrected.

[line 259-261] The authors suggest that asymptomatic should be treated. It would help the reader to highlight how those cases would be identified given that the disease is endemic and the individuals show no symptoms?

43. In that section we were careful to emphasize that “our findings provide a pre-clinical *in vivo* evidence base supporting the concept [of early treatment]”. None of the authors are qualified to make any recommendations about treatment of *T. cruzi*-infected people. Hopefully our data will contribute to considerations about future clinical trials testing the hypothesis. Such trials would be very logistically challenging and, as the reviewer points out, an important barrier would be recruitment of participants who are asymptomatic. Established public health approaches include active case finding or serological surveys.

Figures:

[Figure 1]

a: Is faecal analysis an established read out for constipation?

44. Analysis of stool number, dry and wet weights is commonly reported in studies of constipation. We described the development of the parameters of our assay in our previous paper, including testing levels of faecal retention after increasing lengths of time without access to food, comparing food and water intake, and using fluorescent tracers to analyse stomach emptying and small intestine transit (13). Mice with chronic *T. cruzi*-infection consistently retain significant amounts of faeces in the large intestine in sated and fasted conditions that is not explained by feeding/drinking behaviour or transit dysfunction localising to the stomach or small intestine. We added the key detail to the figure legends that the faecal analysis was done after 4 hours of fasting.

d: Include statistics and perhaps reduce the dot size to better appreciate differences between treatment groups.

45. The original figure 1d was the spleen weights and this did include statistical annotations. We assume the reviewer meant 1g, in which case the changes detailed in point 3 for reviewer 1 should resolve these issues.

[Figure 2]

e. and Figure S1e: Is the faecal weight plotted from individual faecal pellets or has the weight been averaged and normalised to one pellet from a group of pellets?

46. The data points indicate the summed weights of all pellets for each individual animal. The legend has been updated to make this clearer: "pellet weight (sum of all pellets)".

[Figure 3]

a: Can some of these genes up/downregulated in each condition be highlighted on the volcano plots?

47. Yes, some selected genes of interest have now been annotated on the figure.

[Figure 4]

a: Label for the marker (nNOS) in the figure is missing.

48. A label has been added.

d: A GFAP band at 75 kDa has not previously been observed in the gut. Thus, a positive control (such as brain GFAP) should be included to ensure that the antibody is working as expected. In our hands, we never see mouse gut GFAP at 75 kDa. Also, although GFAP degradation has been shown to occur (PMID: 25975510), the resulting band sizes are smaller.

49. We have repeated the analysis with a brain tissue control and saw the expected size for GFAP at 50 kDa. Please refer back to point 27 for further detail on this. We thank the reviewer for bringing the review article on GFAP to our attention. This is now cited where we discuss the Western blot data (Line 255).

e: Figure is not clear.

50. The x axis title now makes clear that the ug refers to the total amount of protein loaded per sample. The figure legend has also been substantially revised to improve clarity.

[Figure 5]

A similar schematic as in Fig. 1a should be included.

51. This has been included in Figure S9.

a: How many control mice were used? This information is missing from figure legend.

52. n = 6, now added to the legend.

Extended data figures

[Figure S1]

Panel h is not clear and needs labelling.

53. We have added some annotation to the H&E micrographs for orientation.

[Figure S2] There is no reference to panel 2d.

54. We now refer to these data at line 127: "There was no correlation between the level of relapse and transit delay in individual mice at discrete time points (Figure S3d), but over time the average levels followed similar, worsening trajectories (Figure S3e)."

[Figure S3] Panel b – how do the changes in Tuj1 mRNA expression reflects the protein levels in figure 4c?

55. Qualitatively, there is not an obvious correlation between the RNA abundance data and the TuJ1 protein visualised in the immunofluorescence images. The RNA was also shown to be significantly increased in BZ-cures in the nanostring data set (see co-ordinates of the Tubb3 gene annotated in Figure 4e), so we are confident that there is genuine upregulation of transcription of this gene after treatment. The discrepancy may reflect the fact that different sample types were used for the two assays – full thickness colon tissue for RNA vs. myenteric plexus only for the IFAs.

[Figure S4] Panel b, Scale bar too close edge

56. This has been fixed.

Methods

Why were only female mice used?

57. We previously compared the course of *T. cruzi* infection in male and female BALB/c mice and found no differences (16). In designing experiments for this study, we aimed to minimise the number of biological variables and given the prior data decided that a single sex study was justified to test the hypothesis. Our funding was awarded on this basis (2018) and did not include the financial resources to meet the costs associated with repeating the experiments in males. We acknowledge that many funders are now moving to approaches where single sex animal studies are not normally considered justified. We have now acknowledged in the abstract that our findings relate to studies in female mice (line 22) and also highlight this at the start of the results (line 89). In future we aim to investigate whether the results we have obtained for females are reproducible in males.

Have the ARRIVE guidelines been followed?

58. Yes, we made every effort to follow them. We have reviewed the manuscript again with reference to the latest ARRIVE 2.0 checklist. The additional details now provided in the methods sections on treatment outcomes (see points 11 and 14) and sample size calculations (see point 16) are in line with best practice recommendations.

REFERENCES

1. PAHO. Guidelines for the diagnosis and treatment of Chagas disease. 2018.
2. Burns AJ, Goldstein AM, Newgreen DF, Stamp L, Schäfer K-H, Metzger M, et al. White paper on guidelines concerning enteric nervous system stem cell therapy for enteric neuropathies. *Dev Biol.* 2016;417(2):229-51.
3. Furness JB. The enteric nervous system and neurogastroenterology. *Nature Reviews Gastroenterology & Hepatology.* 2012;9(5):286-94.
4. Matsuda NM, Miller SM, Evora PR. The chronic gastrointestinal manifestations of Chagas disease. *Clinics (Sao Paulo).* 2009;64(12):1219-24.
5. Soares MB, de Lima RS, Rocha LL, Vasconcelos JF, Rogatto SR, dos Santos RR, et al. Gene expression changes associated with myocarditis and fibrosis in hearts of mice with chronic chagasic cardiomyopathy. *J Infect Dis.* 2010;202(3):416-26.
6. Cunha-Neto E, Dzau VJ, Allen PD, Stamatidou D, Benvenuti L, Higuchi ML, et al. Cardiac gene expression profiling provides evidence for cytokinopathy as a molecular mechanism in Chagas' disease cardiomyopathy. *Am J Pathol.* 2005;167(2):305-13.
7. Ferreira LR, Ferreira FM, Nakaya HI, Deng X, Cândido DD, de Oliveira LC, et al. Blood Gene Signatures of Chagas Cardiomyopathy With or Without Ventricular Dysfunction. *J Infect Dis.* 2017;215(3):387-95.
8. Duque C, So J, Castro-Sesquen YE, DeToy K, Gutierrez Guarnizo SA, Jahanbakhsh F, et al. Immunologic changes are detectable in the peripheral blood transcriptome of clinically asymptomatic Chagas cardiomyopathy patients. *bioRxiv.* 2023.
9. Juiz NA, Torrejón I, Burgos M, Torres AMF, Duffy T, Cayo NM, et al. Alterations in Placental Gene Expression of Pregnant Women with Chronic Chagas Disease. *Am J Pathol.* 2018;188(6):1345-53.
10. de Castro TBR, Canesso MCC, Boroni M, Chame DF, Souza DL, de Toledo NE, et al. Differential Modulation of Mouse Heart Gene Expression by Infection With Two *Trypanosoma cruzi* Strains: A Transcriptome Analysis. *Front Genet.* 2020;11:1031.
11. Faral-Tello P, Greif G, Romero S, Cabrera A, Oviedo C, González T, et al. *Trypanosoma cruzi* Isolates Naturally Adapted to Congenital Transmission Display a Unique Strategy of Transplacental Passage. *Microbiol Spectr.* 2023;11(2):e0250422.
12. Lewis MD, Francisco AF, Taylor MC, Burrell-Saward H, McLatchie AP, Miles MA, Kelly JM. Bioluminescence imaging of chronic *Trypanosoma cruzi* infections reveals tissue-specific parasite dynamics and heart disease in the absence of locally persistent infection. *Cell Microbiol.* 2014;16(9):1285-300.
13. Khan AA, Langston HC, Costa FC, Olmo F, Taylor MC, McCann CJ, et al. Local association of *Trypanosoma cruzi* chronic infection foci and enteric neuropathic lesions at the tissue micro-domain scale. *PLoS Pathog.* 2021;17(8):e1009864.
14. Mann GS, Francisco AF, Jayawardhana S, Taylor MC, Lewis MD, Olmo F, et al. Drug-cured experimental *Trypanosoma cruzi* infections confer long-lasting and cross-strain protection. *PLoS neglected tropical diseases.* 2020;14(4):e0007717.

15. Francisco AF, Jayawardhana S, Taylor MC, Lewis MD, Kelly JM. Assessing the Effectiveness of Curative Benznidazole Treatment in Preventing Chronic Cardiac Pathology in Experimental Models of Chagas Disease. *Antimicrobial agents and chemotherapy*. 2018;62(10).
16. Lewis MD, Francisco AF, Taylor MC, Jayawardhana S, Kelly JM. Host and parasite genetics shape a link between *Trypanosoma cruzi* infection dynamics and chronic cardiomyopathy. *Cell Microbiol*. 2016;18(10):1429-43.
17. Francisco AF, Lewis MD, Jayawardhana S, Taylor MC, Chatelain E, Kelly JM. The limited ability of posaconazole to cure both acute and chronic *Trypanosoma cruzi* infections revealed by highly sensitive in vivo imaging. *Antimicrobial agents and chemotherapy*. 2015;59(8):4653-61.
18. Santi-Rocca J, Fernandez-Cortes F, Chillón-Marinas C, González-Rubio M-L, Martin D, Gironès N, Fresno M. A multi-parametric analysis of *Trypanosoma cruzi* infection: common pathophysiologic patterns beyond extreme heterogeneity of host responses. *Scientific Reports*. 2017;7(1):1-12.
19. Silberstein E, Serna C, Fragoso SP, Nagarkatti R, Debrabant A. A novel nanoluciferase-based system to monitor *Trypanosoma cruzi* infection in mice by bioluminescence imaging. *PLOS ONE*. 2018;13(4):e0195879.
20. Khare S, Nagle AS, Biggart A, Lai YH, Liang F, Davis LC, et al. Proteasome inhibition for treatment of leishmaniasis, Chagas disease and sleeping sickness. *Nature*. 2016;537(7619):229-33.
21. Hossain E, Khanam S, Dean DA, Wu C, Lostracco-Johnson S, Thomas D, et al. Mapping of host-parasite-microbiome interactions reveals metabolic determinants of tropism and tolerance in Chagas disease. *Science advances*. 2020;6(30):eaaz2015.
22. Calvet CM, Silva TA, Thomas D, Suzuki B, Hirata K, Siqueira-Neto JL, McKerrow JH. Long term follow-up of *Trypanosoma cruzi* infection and Chagas disease manifestations in mice treated with benznidazole or posaconazole. *PLoS neglected tropical diseases*. 2020;14(9):e0008726.
23. Wesley M, Moraes A, Rosa AC, Lott Carvalho J, Shiroma T, Vital T, et al. Correlation of Parasite Burden, kDNA Integration, Autoreactive Antibodies, and Cytokine Pattern in the Pathophysiology of Chagas Disease. *Front Microbiol*. 2019;10:1856.
24. Ward AI, Lewis MD, Khan AA, McCann CJ, Francisco AF, Jayawardhana S, et al. *In Vivo* Analysis of *Trypanosoma cruzi* Persistence Foci at Single-Cell Resolution. *mBio*. 2020;11(4):e01242-20.
25. Lewis MD, Francisco AF, Jayawardhana S, Langston H, Taylor MC, Kelly JM. Imaging the development of chronic Chagas disease after oral transmission. *Scientific Reports*. 2018;8(1):11292.
26. Alonso-Vega C, Urbina JA, Sanz S, Pinazo MJ, Pinto JJ, Gonzalez VR, et al. New chemotherapy regimens and biomarkers for Chagas disease: the rationale and design of the TESEO study, an open-label, randomised, prospective, phase-2 clinical trial in the Plurinational State of Bolivia. *BMJ Open*. 2021;11(12):e052897.
27. Torrico F, Gascon J, Ortiz L, Alonso-Vega C, Pinazo MJ, Schijman A, et al. Treatment of adult chronic indeterminate Chagas disease with benznidazole and three E1224 dosing regimens: a proof-of-concept, randomised, placebo-controlled trial. *Lancet Infect Dis*. 2018;18(4):419-30.
28. Lambert ML, Hasker E, Van Deun A, Roberfroid D, Boelaert M, Van der Stuyft P. Recurrence in tuberculosis: relapse or reinfection? *Lancet Infect Dis*. 2003;3(5):282-7.
29. Goyal V, Das VNR, Singh SN, Singh RS, Pandey K, Verma N, et al. Long-term incidence of relapse and post-kala-azar dermal leishmaniasis after three different visceral leishmaniasis treatment regimens in Bihar, India. *PLoS neglected tropical diseases*. 2020;14(7):e0008429.
30. Kulkarni S, Micci M-A, Leser J, Shin C, Tang S-C, Fu Y-Y, et al. Adult enteric nervous system in health is maintained by a dynamic balance between neuronal apoptosis and neurogenesis. *PNAS*. 2017;114(18):E3709-E18.
31. Belkind-Gerson J, Carreon-Rodriguez A, Benedict LA, Steiger C, Pieretti A, Nagy N, et al. Nestin-expressing cells in the gut give rise to enteric neurons and glial cells. *Neurogastroenterology & Motility*. 2013;25(1):61-e7.
32. Laddach A, Chng SH, Lasrado R, Progzatzky F, Shapiro M, Erickson A, et al. A branching model of lineage differentiation underpinning the neurogenic potential of enteric glia. *Nat Commun*. 2023;14(1):5904.
33. Torrico F, Gascón J, Barreira F, Blum B, Almeida IC, Alonso-Vega C, et al. New regimens of benznidazole monotherapy and in combination with fosravuconazole for treatment of Chagas disease (BENDITA): a phase 2, double-blind, randomised trial. *Lancet Infect Dis*. 2021;21(8):1129-40.

REVIEWERS' COMMENTS

Reviewer #1 (Remarks to the Author):

The authors have adequately responded to my concerns.

Rev. 1

Reviewer #3 (Remarks to the Author):

The authors have made a great effort to address the concerns and comments raised by the reviewer by performing additional experiments. The new data has significantly strengthened the manuscript. The reviewer would like to suggest a minor comment that is important to clearly interpret the new data and confirm conclusions. The additional assays demonstrating neuronal cell death (TUNEL, caspase-3) and glial proliferation (EdU) are important new data that further support neuronal cell death after infection and proliferative activation of glial cells, however, the images are not clearly visible in its current format. Apart from this minor comment, the reviewer is satisfied with the response of the authors and the changes they performed.

Reviewer #4 (Remarks to the Author):

I was asked to review the comments of reviewer 2 and the authors responses as reviewer 2 was unable to do so.

In preparation for doing that I read the revised version, the comments and the author responses. In my opinion, the paper is an interesting one which makes several strong contributions to an under-researched but important area and is suitable for publication.

Reviewer 2 provided a strong critical review of the paper and highlighted several weaknesses for the author to address and several areas for clarification.

The authors have been thorough in their response, working point by point through the critique and addressing most of the major points raised experimentally and analytically and revising the work appropriately, improving those aspects of the manuscript.

Those aspects that they chose to rebut rather than amend were primarily semantic - I can see why the reviewer raised them and why the author has declined to revise as suggested by the reviewer.

From my perspective I am happy for the author to have the discretion as their preferred terminology isn't grossly inaccurate and doesn't substantially affect the paper. In general the authors have done a good job with the revisions and the paper is improved by them.